# Ancient DNA study reveals HLA susceptibility locus for leprosy in medieval Europeans

Ben Krause-Kyora [1,2], Marcel Nutsua[1], Lisa Boehme[1], Federica Pierini[3], Dorthe Dangvard Pedersen [4], Sabin-Christin Kornell[1], Dmitriy Drichel[5], Marion Bonazzi[1], Lena Möbus[1], Peter Tarp[4], Julian Susat[1], Esther Bosse[1], Beatrix Willburger[6], Alexander H. Schmidt[6], Jürgen Sauter[6], Andre Franke [1], Michael Wittig[1], Amke Caliebe[7], Michael Nothnagel [5], Stefan Schreiber[1,8], Jesper L. Boldsen[4], Tobias L. Lenz[3] & Almut Nebel[1]

Leprosy, a chronic infectious disease caused by *Mycobacterium leprae* (*M. leprae*), was very common in Europe till the 16th century. Here, we perform an ancient DNA study on medieval skeletons from Denmark that show lesions specific for lepromatous leprosy (LL). First, we test the remains for *M. leprae* DNA to confirm the infection status of the individuals and to assess the bacterial diversity. We assemble 10 complete *M. leprae* genomes that all differ from each other. Second, we evaluate whether the human leukocyte antigen allele DRB1*15:01, a strong LL susceptibility factor in modern populations, also predisposed medieval Europeans to the disease. The comparison of genotype data from 69 *M. leprae* DNA-positive LL cases with those from contemporary and medieval controls reveals a statistically significant association in both instances. In addition, we observe that DRB1*15:01 co-occurs with DQB1*06:02 on a haplotype that is a strong risk factor for inflammatory diseases today.

[1] Institute of Clinical Molecular Biology, Kiel University, Kiel 24105, Germany. [2] Max Planck Institute for the Science of Human History, Jena 07745, Germany. [3] Department of Evolutionary Ecology, Research Group for Evolutionary Immunogenomics, Max Planck Institute for Evolutionary Biology, Plön 24306, Germany. [4] Department of Forensic Medicine, Unit of Anthropology (ADBOU), University of Southern Denmark, Odense S 5260, Denmark. [5] Department of Statistical Genetics and Bioinformatics, Cologne Center for Genomics (CCG), University of Cologne, Cologne 50931, Germany. [6] DKMS, Tübingen 72072, Germany. [7] Institute of Medical Informatics and Statistics, Kiel University, Kiel 24105, Germany. [8] Clinic for Internal Medicine, University Hospital of Schleswig-Holstein, Kiel 24105, Germany. These authors contributed equally: Marcel Nutsua, Lisa Boehme, Federica Pierini, Dorthe Dangvard Pedersen, Sabin-Christin Kornell. These authors jointly supervised this work: Jesper L. Boldsen, Tobias L. Lenz, Almut Nebel. Correspondence and requests for materials should be addressed to B.K.-K. (email: b.krause-kyora@ikmb.uni-kiel.de)

Leprosy is a chronic infectious disease caused by *Myco-bacterium leprae* (*M. leprae*). It was very common during the Middle Ages in Europe, from where it almost completely disappeared in the 16th century[1]. A recent ancient DNA (aDNA) analysis has revealed a high level of *M. leprae* genome conservation over the past 1000 years, indicating that the leprosy epidemic during the European Middle Ages was unlikely to be due to particularly virulent strains[2]. Instead, other factors such as malnutrition, co-infections, and host genetics may have increased disease susceptibility in the medieval period. At present, leprosy is virtually absent in Europe, but still remains a big health problem in South-East Asia (e.g., India), North and Central Africa (Central African Republic, Democratic Republic of the Congo), Oceania (Indonesia, Papua New Guinea) and the Americas (Brazil, Mexico)[3]. All 10 modern human *M. leprae* genomes sequenced up to now fall in five distinct phylogenetic branches that show a specific geographic distribution pattern[2].

In addition to environmental factors, predisposition to leprosy is considerably influenced by variation in immune-related genes[4]. Very strong disease associations were reported for the human leukocyte antigen (HLA) region[4–6]. Although the involvement of both HLA class I and class II genes was intensively studied in leprosy[6], class II alleles, particularly in the DRB1 locus, were shown to be most consistently associated with the disease[5–8]. Of these, DRB1*15:01 is the most notable risk factor for lepromatous leprosy (LL) in India, China, and Brazil today[7–9]. LL represents a severe form of the disease, characterized by a high bacterial load and specific lesions that can reliably be diagnosed on bones[1,10]. DRB1*15:01 influences LL susceptibility in very diverse populations, therefore, it remains an interesting open question whether the same association also existed in Europeans of the Middle Ages. We tested this hypothesis by an aDNA analysis in medieval skeletons with signs of LL.

Here, we isolated aDNA from remains with LL lesions from one locale, the St. Jørgen leprosarium in Odense, Denmark[11] (historically dated 1270–1550 AD; Fig. 1). First, we analyzed extracts for the presence of *M. leprae* DNA to confirm the infection status of the individuals and to assess the bacterial diversity. We assembled 10 complete *M. leprae* genomes that all differ from each other and represent three different branches. Second, we performed an aDNA association study to evaluate whether the risk allele DRB1*15:01 predisposed medieval Europeans to LL. The comparison of genotype data from 69 *M. leprae* DNA-positive cases with those from contemporary and medieval controls revealed a statistically significant association in both instances. In addition, we studied the DRB1 locus and its haplotype structure in more detail in the LL cases. We observed that DRB1*15:01 co-occurred on a haplotype that also carried DQB1*06:02; this haplotype is a strong risk factor for inflammatory diseases in present-day populations.

## Results

**Ancient association study with LL**. DNA extracts from 85 specimens with LL-specific bone lesions from the St. Jørgen cemetery were investigated for the DRB1*15:01 allele (Supplementary Note 1). Direct Sanger sequencing of the DRB1 exon is not possible from aDNA, because the length of the relevant exon coding for the antigen-binding domain far exceeds the average DNA fragment size of ancient samples and the polymorphic exon sequence precludes the use of intra-exon primers. However, the SNP allele rs3135388-T is an established marker for high-throughput genotyping of DRB1*15:01 in disease studies[12,13]. Genotyping a single SNP requires only a very short amplicon size, which renders this marker much more suitable for analysis of the highly fragmented aDNA. When we performed PCR-based

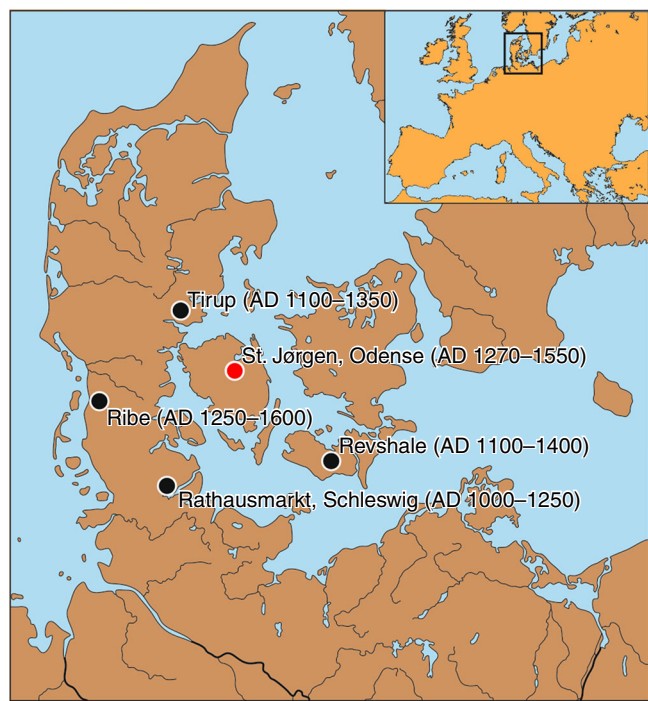

**Fig. 1** Origin of medieval samples. Geographic location of the medieval cemeteries in southern Denmark and northern Germany from where the specimens in this study were obtained. Dates in parentheses indicate the time span of active use as cemeteries. Red dot marks the cemetery with LL-positive samples. LL: lepromatous leprosy. The software CorelDRAW was used to create the map (designed by B.K.-K.)

Sanger sequencing for the SNP allele in the 85 St. Jørgen specimens, 69 of them contained sufficient endogenous nuclear DNA to yield genotype data (Supplementary Data 1, Supplementary Note 2, Supplementary Fig. 1, Supplementary Table 1). These 69 LL samples also tested positive for the presence of *M. leprae* DNA by shotgun high-throughput sequencing (HTS) (see below and Supplementary Table 2, 3) and/or screening PCR (Supplementary Data 1).

We first performed an association test for rs3135388 and medieval leprosy considering the 69 *M. leprae*-positive LL individuals from St. Jørgen as the case group and a large sample of contemporary northern Germans as controls, using DRB1 genotype information from an extensive bone marrow database. The rs3135388-T allele frequency in the LL individuals (0.283) was significantly higher than in the controls (0.138; $p = 9.49*10^{-06}$, OR = 2.46; two-sided Fisher's exact test, Fig. 2, Table 1). Subsequently, we compared the rs3135388-T frequency in the 69 St. Jørgen individuals with genetic information from those in medieval controls. Such a case–control study would require—besides the herein-studied 138 alleles of our cases—another 250 alleles in controls to obtain an a priori power of 75% (see Methods). We were able to generate genotype data from 152 randomly sampled individuals (i.e., 304 alleles) excavated from four medieval sites in Denmark and northern Germany (Fig. 1, Table 1, Supplementary Note 1). Although the rs3135388-T frequencies did not differ significantly between the four cemeteries ($p = 0.932$, two-sided Fisher's exact test), LL cases from St. Jørgen showed a significant enrichment of this allele compared to the combined set of medieval controls (0.283 vs. 0.184; $p = 0.024$; OR = 1.74; two-sided Fisher's exact test, Table 1, Fig. 2).

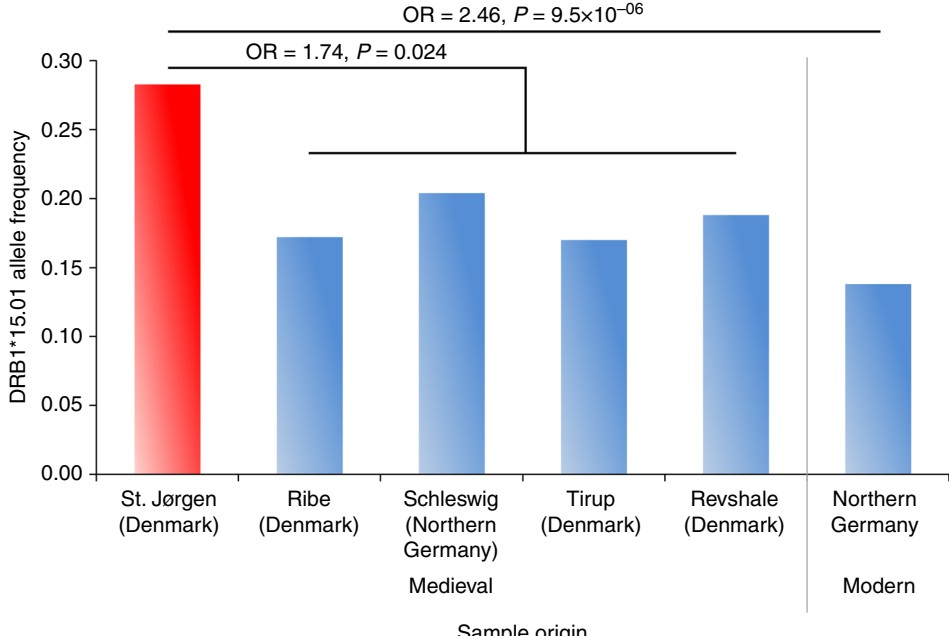

**Fig. 2** Frequency of rs3135388-T variant. Allele frequency of rs3135388-T in 69 LL-positive medieval St. Jørgen individuals (red bar) and in four sample sets from medieval unaffected controls, compared with DRB1*15:01 data in a contemporary cohort of northern Germany from the DKMS bone marrow database (see Table 1 and Supplementary Information for sample sizes and other details). Odds ratio (OR) and P value from Fisher's exact test are reported. LL: lepromatous leprosy

**M. leprae genome analysis and metagenomic screening**. DNA extracts from 68 of the 69 specimens with LL-specific bone lesions were successfully subjected to HTS, without any prior *M. leprae* or human genome enrichment (Supplementary Table 4). Damage patterns of both human and *M. leprae* reads were consistent with an ancient origin of the DNA (Supplementary Note 2, 3, Supplementary Table 2, 5–9). Metagenomic analysis of the HTS reads using MALT[14] showed the typical spectrum of soil bacteria and no signs of bacterial co-infection (Supplementary Note 3, Supplementary Fig. 2, 3). For ten specimens, the *M. leprae* genome coverage was sufficient to allow for de novo assemblies of the HTS data without any enrichment bias (Supplementary Note 3, Supplementary Table 2, 10-13). Genome-wide comparisons and SNP effect analysis did not show any variants that would indicate a change in virulence or function in these ten genomes (Supplementary Note 3, Supplementary Data 2). We then included the ten well-covered *M. leprae* genomes together with another genome from the St. Jørgen cemetery (Jørgen_625), four additional medieval genomes and 12 modern samples, all published elsewhere[2], in a phylogenetic analysis. The eleven genomes from St. Jørgen fell into 3 branches (Fig. 3, Supplementary Note 4, Supplementary Fig. 4–7, Supplementary Data 3).

**HTS-based HLA typing**. Given the association between LL and rs3135388-T, which is known to be statistically associated with the presence of DRB1*15:01 in Europeans, we subsequently focused on the HLA class II region in more detail in the medieval LL individuals. We adapted an existing DNA capture[15] method and optimized it for short aDNA fragments. After HTS, we called HLA alleles using a new aDNA-optimized analysis pipeline (Supplementary Fig. 8). Depending on each sample's aDNA quality and resulting sequence coverage, HLA alleles were called at two different levels of resolution, following the official HLA nomenclature[16]: The broader two-digit (now also called 1st-field) resolution defines different functional lineages of alleles, yielding a level of separation largely equivalent to the classical serotyping

used in early transplantation medicine (e.g., DRB1*15 vs. DRB1*01). The finer four-digit (now also called 2nd field) resolution separates alleles that share functional properties but differ in their protein sequence (e.g., DRB1*15:01 vs. DRB1*15:02, differing in 1 out of 89 codons of the molecule's antigen-binding groove). Using our target capture and HTS approach, we were able to determine two- or four-digit resolution for HLA class II genes (Supplementary Table 14, 15). Of the 136 alleles investigated (= 68 screened individuals), we succeeded in calling 82 DRB1 alleles at the two-digit level and of these, 57 alleles at the four-digit resolution (Supplementary Table 14). In some samples, the DNA quality and thus read coverage did not allow for more precise allele calls. Among the 82 DRB1 alleles at two-digit resolution, 20 were determined to belong to the DRB1*15 lineage. In modern Europeans, by far the most common DRB1*15 molecule variant is DRB1*15:01 (allele frequency in northern Germany: 0.138, compared with 0.007 of the second most common allele *15:02), rendering it highly likely that those DRB1*15 calls represent the allele DRB1*15:01. This was confirmed as 13 of the 20 DRB1*15 alleles could also be called at four-digit resolution, in each case showing DRB1*15:01. If all 20 DRB1*15 alleles were to consist of the DRB1*15:01 allele, this would result in a frequency of 0.24, very similar to 0.28 as measured by rs3135388 in the 69 individuals. The deviation between the two frequency estimates is owing to the much smaller number of LL cases that could successfully be genotyped by HLA sequencing compared with SNP genotyping. In the 43 LL samples for which allele calls were available from both PCR- and HTS-based methods, we confirmed that rs3135388-T always co-occurred with DRB1*15:01 ($r^2$ = 1.0) (Supplementary Table 15, Supplementary Data 1), verifying the suitability of the T allele for detecting the presence of DRB1*15:01. Interestingly, in all carriers of DRB1*15:01 (representing 12 individuals, including one homozygote), we also observed the allele DQB1*06:02. The same was true in reverse, i.e., all individuals for which DQB1*06:02 could be called (n = 20) also had either DRB1*15:01, DRB1*15 (not

**Table 1 Frequency of the rs3135388-T allele**

| | LL cases | LL controls | | | | |
|---|---|---|---|---|---|---|
| | | Medieval | | | | Modern |
| Site | Odense, St. Jørgen (Denmark) | Ribe (Denmark) | Schleswig, Rathausmarkt (northern Germany) | Tirup (Denmark) | Revshale (Denmark) | Northern Germany* modern |
| Dating | 1270–1550 AD | 1250–1600 AD | 1000–1250 AD | 1100–1350 AD | 1100–1400 AD | modern |
| Number of individuals | 69 | 32 | 49 | 47 | 24 | 129,336 |
| Number of rs3135388-T alleles | 39 | 11 | 20 | 16 | 9 | 35,681 |
| Allele frequency | 0.283 | 0.172 | 0.204 | 0.170 | 0.188 | 0.138 |

LL lepromatous leprosy. *HLA DRB1*15:01 data provided by DKMS (Supplementary Note 6)

allowing for more precise allele call), or had an incomplete allele call (because of insufficient aDNA quality). This observation suggests strong linkage disequilibrium (LD) between the two loci (Supplementary Note 5, Supplementary Table 15). The two alleles indeed define a DRB1-DQB1 haplotype that is still found in modern Europeans at a considerable frequency (13.3%, provided by DKMS, Supplementary Note 6).

**Binding properties of DRB1 alleles.** As the binding and presentation of antigenic peptides to immune effector cells is the key function of HLA molecules, we also explored the relative binding properties of the detected HLA alleles[17] with regard to potential *M. leprae* antigens. We found that among 18 contemporarily common DRB1 alleles, DRB1*15:01 is predicted to bind the second-smallest number of potential *M. leprae* antigens (DRB1*15:01: 11 out of 5345 peptides, mean of 18 DRB1 alleles: 64.7; Fig. 4). HLA binding-prediction for the entire *M. leprae* proteome (516,303 unique peptides) still revealed limited relative presentation capacity for DRB1*15:01, but to a lesser extent (Supplementary Fig. 9). The fact that peptide binding of DRB1*15:01 is relatively more limited when focusing on potential antigenic proteins suggests that it might be particularly ineffective in the context of antigen presentation. Limited antigen presentation could impair specific immunity against *M. leprae* infections and thus confer susceptibility to its carriers, which is exactly what we found in the association analysis above.

## Discussion

Leprosy was endemic during the Middle Ages in Europe, where it reached its greatest prevalence between AD 1200 and 1400[1]. The disease was greatly feared because it caused visible disfigurement, was incurable and contagious. Stigmatization and social segregation of patients were common in the medieval period; especially those affected with the severe form of LL were quarantined in so-called leprosaria or "leper houses". Deceased LL patients were buried in special leprosaria-associated cemeteries as, for instance, in St. Jørgen in the Danish town of Odense[10,11]. In the present aDNA study, we successfully analyzed 69 human remains from St. Jørgen. All specimens showed typical LL bone lesions and were *M. leprae* DNA-positive.

From 10 samples with the best *M. leprae* DNA content, complete bacterial genomes were generated without any enrichment, which—together with a previously published one[2]—resulted in a total of eleven genomes from St. Jørgen alone. This high success rate is remarkable and could be due to the very good preservation of *M. leprae* DNA in general[2] or certain favorable environmental conditions at this site. It could also be a consequence of the high bacterial load associated with the severe LL form of leprosy[18]. Of note, the number of medieval *M. leprae* genomes published so far ($n = 15$, including this study) exceeds that of modern bacteria

and has led to a considerable increase in the phylogenetic and temporal resolution. The 11 St. Jørgen genomes differed from each other and fell into three branches (Fig. 3, Supplementary Note 3, 4, Supplementary Fig. 4–7). In our relatively small data set, we did not observe any correlation between *M. leprae* strains and the HLA allele DRB*15:01. We identified an ancient strain in branch 0 with the SNP type 3K. This type has previously been reported in medieval Hungarians and Byzantinians[19,20]; today it shows a wide geographic distribution ranging from the Near East to Oceania[2,19,20]. Furthermore, 10 of the 11 St. Jørgen genomes clustered in branches 2 and 3, representing two types that were already described in skeletons from the European Middle Ages[2,18,19]. Among the St. Jørgen samples, branch 3-strains were most abundant ($n = 9$) and very similar to extant strains recently identified in European red squirrels[21] (Fig. 3). Interestingly, modern branch 3-bacteria have the ability to infect at least three different hosts including humans and squirrels[21]. The presence of so many different strains in one branch and in a rather remote locale over a few hundred years is surprising. In view of the very low mutation rate of the pathogen and the relatively short time window, one would expect more similar, even identical, genomes. This interesting observation indicates multiple sources of infection with different strains. In agreement with previous observations[2,22], no variants in the *M. leprae* genomes were identified that would suggest a change in bacterial virulence or function. In particular, there were no apparent changes in the outer membrane peptides that could lead to different binding affinities to HLA class II proteins. This finding highlights the importance of investigating host genetic factors to explain the high disease prevalence in the medieval period.

As the HLA allele DRB1*15:01 is the most notable risk factor for LL in various modern populations worldwide[4,5,7,8], we wanted to test the hypothesis whether this HLA allele also predisposed Europeans to the disease during the medieval leprosy epidemic. To this end, we conducted a two-stage association analysis for SNP rs3135388. Its T allele is an established and reliable marker for DRB1*15:01[12,13]. First, we compared rs3135388 data from 69 St. Jørgen LL samples, which were all *M. leprae* DNA-positive, with those from contemporary controls. Such an analysis between medieval and modern allele frequencies is valid under the assumption that the common genetic make-up of the Danish population has not significantly changed since the 11th century. This assumption is corroborated by the observation that those 53 St. Jørgen samples that were of sufficient data quality fell within the variability of modern northern Europeans (Fig. 5, Supplementary Note 7). In addition, mitochondrial DNA (mtDNA) haplogroup frequencies at St. Jørgen were comparable to those of northern Europeans today (Supplementary Data 1). These findings indicate no major genome-wide changes in the Danish population structure in the past 1000 years. Furthermore, based

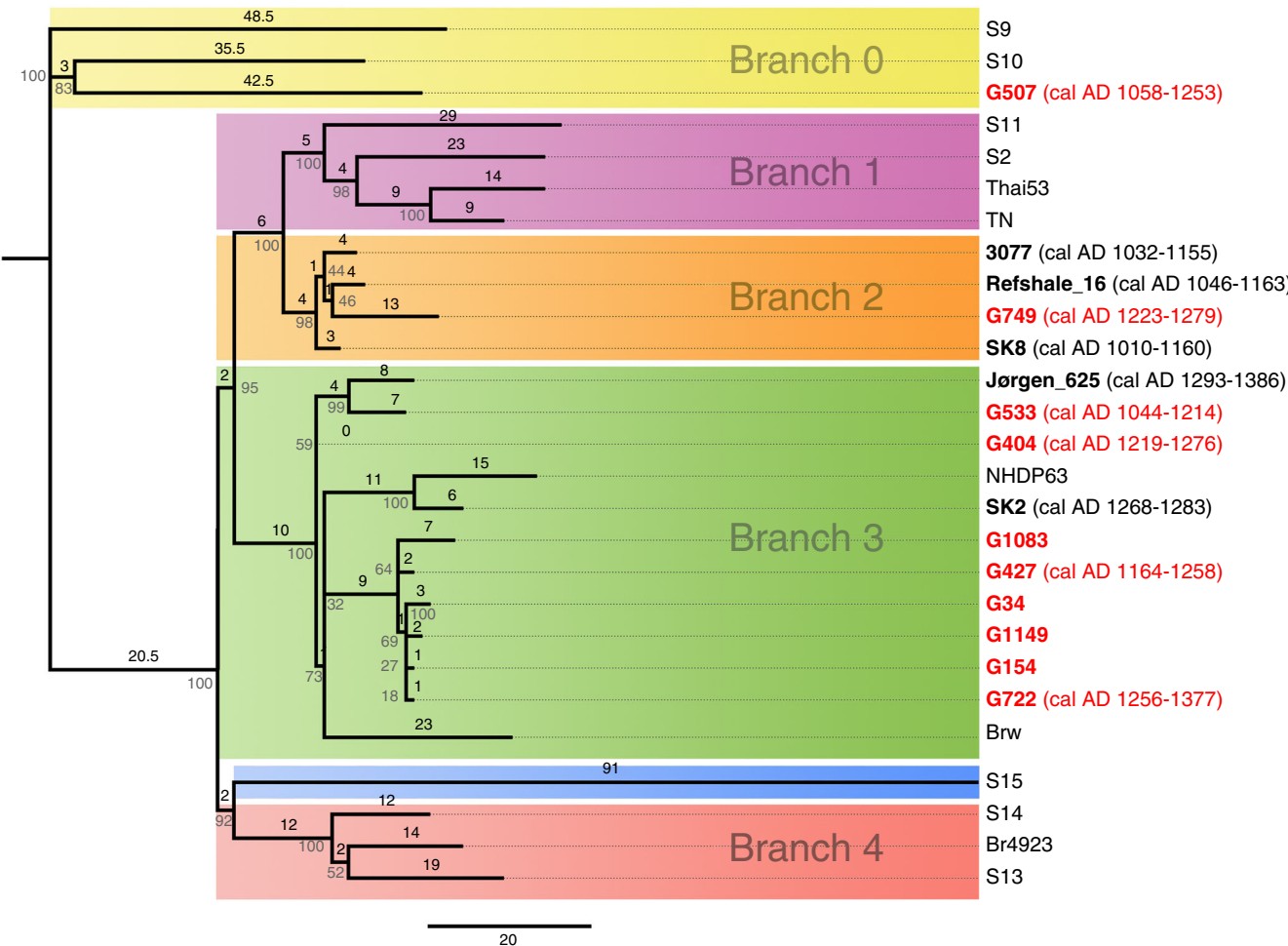

**Fig. 3** Phylogeny of medieval and extant *M. leprae* strains. Phylogeny of medieval and modern *M. leprae* genomes using a maximum parsimony tree. Labeling colors: red bold (ancient genomes generated in this study), black bold (ancient genomes generated in a previous study (2)) and black (modern genomes generated in previous studies (2, 13)). Calibrated radiocarbon dates (2 sigma range) given in parentheses refer to branch tips. Bootstrap node support (in percent, from 500 replicates) is shown in gray numbers next to the branches, whereas the number of nucleotide substitutions on each branch is set in black. Color shading highlights different phylogenetic branches

on estimates for identical-by-decent (IBD) sharing we determined that the St. Jørgen individuals were not directly related to each other (Supplementary Data 4, Supplementary Table 16). The comparison of the rs3135388-T frequencies from the 69 LL cases with the DRB1*15:01 data from modern controls revealed a statistically significant association. To strengthen this finding, we further analyzed whether the allele frequency in the cases was different from that of medieval controls. The controls (i) were selected from sites that were geographically close and dated earlier or contemporaneous to St. Jørgen, (ii) showed no osteological evidence of the severe LL form of the disease and (iii) were *M. leprae* DNA-negative (Supplementary Data 1). The rs3135388-T frequency differed significantly between ancient cases and controls with an odds ratio similar in size to previous reports about DRB1*15:01[4,5]. Notably, the medieval control frequency was slightly but significantly higher than today (*p* = 0.024, OR = 1.41, Table 1), possibly indicating weak selection against this allele during or since medieval times. This observation raises the question of how leprosy might have led to reduced reproductive fitness given that affected people, at least in modern populations, rarely die from the disease. Leprosy patients in the Middle Ages had to endure rejection and isolation in leprosaria and were not allowed to marry[23]. In addition, leprosy is known today to result in a hormone-related decrease in fertility[24,25] and a higher

vulnerability to other infections[26–29]. By contrast, the fact that this allele is still the most common DRB1 allele in contemporary northern Germany suggests that it is (and was) associated with further and likely antagonistic fitness effects that prevented stronger frequency declines in medieval times.

Taken together, our results demonstrate a significant association between the HLA class II region and LL susceptibility in medieval leprosy patients from northern Europe, involving the DRB1*15:01 allele. As this allele is predicted to bind only a very small number of potential *M. leprae* antigens, this observation lends support to the hypothesis that limited HLA-presentation of relevant antigens may have impaired an *M. leprae*-specific immune response, leading to increased susceptibility to LL. One could speculate that, since DRB1*15:01 apparently was one of the most common DRB1 alleles in medieval northern Europe (or at least in Denmark, according to the control samples), *M. leprae* proteins might have evolved to evade presentation by this common allele, following a process of negative frequency-dependent selection. However, further molecular data and comparative work is required to address this hypothesis.

Interestingly, the DRB1*15:01-DQB1*06:02 haplotype represents an allele combination that is still common in contemporary Europeans[30,31]. Furthermore, in modern populations, it is a strong risk factor for ulcerative colitis, sarcoidosis, and multiple

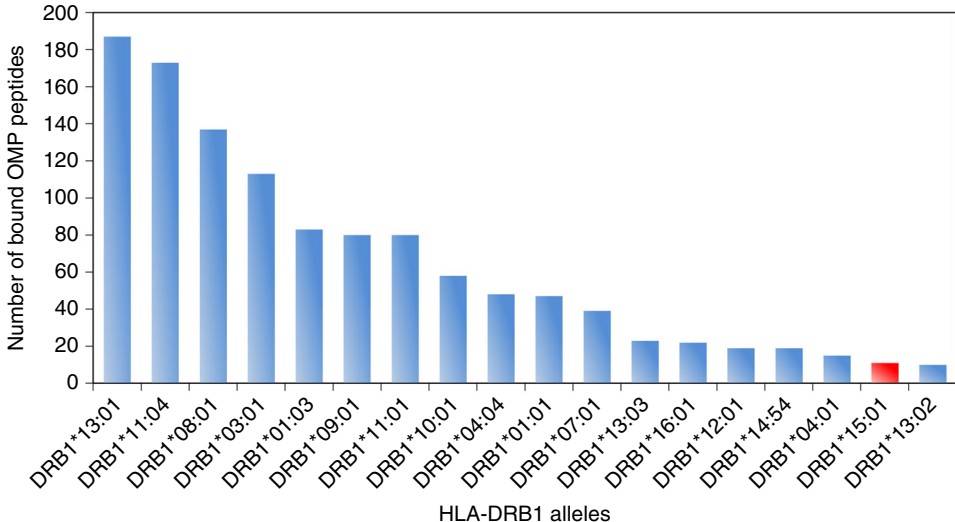

**Fig. 4** Functional association between DRB1*15:01 and *M. leprae*. Computational prediction of HLA-presented antigenic *M. leprae* peptides for common HLA-DRB1 alleles revealed that DRB1*15:01 (red bar) presents one of the smallest *M. leprae* antigen repertoires. Binding predictions were run for the 18 HLA-DRB1 alleles with an allele frequency of >1% in representative contemporary samples from Schleswig-Holstein/Germany n = 129,336) and a Danish minority population from northern Germany (n = 918). OMP—outer membrane proteins

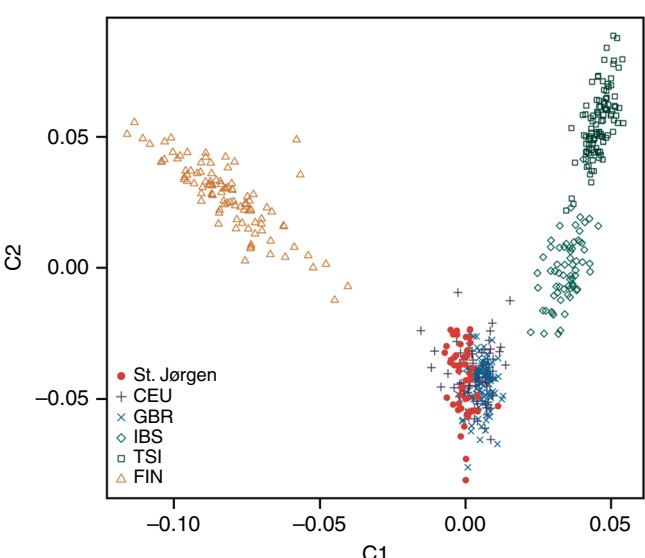

**Fig. 5** Relationship of 53 medieval leprosy-positive Danes to contemporary Europeans. Principal component analysis plot for 53 medieval St. Jørgen individuals in relation to European population samples from the 1000 Genomes project. (CEU, Northern Europeans from Utah; GBR, British in England and Scotland; IBS, Iberian population in Spain; TSI, Tuscans in Italy; FIN, Finnish in Finland)

sclerosis[32–34], whereas being protective against type-1 diabetes[35]. These different disease associations highlight the well-known pleiotropy of HLA variants that affect the population frequency of specific haplotypes and contribute to the genetic diversity in the HLA region in general. More generally, our findings provide a new, temporal layer of evidence for the hypothesis that ancient epidemics such as leprosy have influenced the present-day frequency of genetic factors associated with modern inflammatory diseases[36,37].

## Methods
**Selection of archeological specimens**. In the current study, human skeletal remains were obtained from five medieval cemeteries in Denmark and northern

Germany (Fig. 1, Supplementary Note 1). The specimens were analyzed with permission from the respective museums and collections (Horsens Museum, Museum Lolland-Falster, Odense Bys Museer, Sydvestjyske Museer, Arch-äologisches Landesmuseum der Stiftung Schleswig Holsteinische Landesmuseen Schloss Gottorf). Osteological information (sex, age at death, leprosy status) of all individuals was collected. Individuals who suffered from LL and were buried at the leprosarium cemetery of St. Jørgen in Odense are considered cases in our study. Given the high infection prevalence, people interred at ordinary cemeteries might also have been infected. However, if this was the case, they were affected with a milder form of the disease, as evidenced from osteological analysis. These individuals were here regarded as controls. They were selected from the four sites Ribe, Revshale, Tirup, and Schleswig/Rathausmarkt that were geographically close and dated earlier or contemporaneous to Odense/St. Jørgen.

**Sample processing**. Material from each selected skeleton was collected in person by a member of the Kiel aDNA group or a member of the Odense ADBOU group to minimize the risks of contamination. Two different types of samples were taken: teeth and/or petrous bones (Supplementary Data 1). The DNA extractions and pre-PCR steps were carried out in clean room facilities dedicated to aDNA research. Following the guidelines on contamination control in aDNA studies[38–40], all surfaces and re-usable utensils were extensively cleaned with bleach before and after work. Besides, UV lights were used to improve the decontamination process. In addition, negative controls were included in each step involving non-indexed DNA molecules. Finally, access of staff and objects to the laboratories was restricted to a one-direction route from the pre-PCR rooms to the post-PCR facility. The post-PCR room was located in an independent building. This room was used to run the screening and indexing PCRs (the reaction mixes were prepared beforehand in the dedicated pre-PCR rooms) and to process the indexed PCR products (purification, amplification). As for the pre-PCR laboratories, no modern leprosy studies had ever been conducted there prior to the start of this project.

Whole teeth were cleaned in pure bleach solution (sodium hypochlorite) and rinsed with purified water. After drying at 37 °C overnight, the samples were ground in a ball mill homogenizer for 45 s at maximum speed. Petrous bones were cleaned in pure bleach solution (sodium hypochlorite), rinsed with purified water and dried overnight at 37 °C. The inner ear (cochlea and vestibule) was cut out with a bone saw. Subsequently, the cleaned (with bleach analogous to the petrous bone) and dried inner ear piece was ground in a ball mill homogenizer for 45 s at maximum speed.

**DNA extraction for PCR assays**. After grinding, 50 mg of bone powder were incubated in 500 μL ethylenediaminetetraacetic acid (EDTA; pH 8, 0.5 M) and 20 μL proteinase K (0.25 mg/mL) under gentle rotation at 37 °C overnight. The suspension was centrifuged for 3 min at 6000 rpm and 200 μL of the supernatant were used for DNA extraction with the Qiagen EZ1 Advanced Investigator Kit (program setting "Trace"). Purified DNA was eluted in 50 μL TE buffer and stored at −20 °C until further use. One negative control was processed per five samples[41].

**DNA extraction for HTS**. A total of 50 mg of bone powder were incubated in 960 μL EDTA (pH 8, 0.5 M) and 40 μL proteinase K (0.25 mg/mL) under gentle

rotation at 37 °C overnight. The suspension was further processed according to a published protocol by Dabney et al.[42].

**PCR-based experiments**. For this study, the following three PCR-based setups were carried out: *M. leprae* DNA screening (RLEP and 18-kDA), nuclear DNA SNP rs3135388 and mtDNA (hypervariable regions I and II) analysis. PCRs were performed in a 25-μL volume containing 1× Immolase buffer (Bioline), 0.04 U/μL Immolase DNA Polymerase (Bioline), 1.5 mM MgCl2 (Bioline), 4% DMSO, 200 μM dNTP mix (Bioline) and 0.4 μM of each primer (see Supplementary Table 1 and Supplementary Fig. 1 for primer and amplicon details). For the *M. leprae* and the SNP screening, 5 μL of aDNA extract were used as template. For the mtDNA PCR, the amount of extract added was 1 μL. The annealing temperatures were 52 °C (*M. leprae* DNA screening), 58 °C (SNP PCR) or 60 °C (mtDNA screening), respectively. PCR success was evaluated by gel electrophoresis. Subsequent Sanger sequencing was performed following standard procedures.

**PCR-based *M. leprae* DNA screening**. All 85 samples from the St. Jørgen cemetery and all 223 controls were subjected to PCR-based *M. leprae* DNA screening[2,18]. Two primer pairs were used to amplify regions coding for RLEP (130 bp) and the 18-kDa protein (98 bp). A sample was considered positive for *M. leprae* when at least one PCR yielded a product of the expected length and sequence. If the amplicon did not match the expected sequence, a nucleotide BLAST was performed to identify possible contamination sources.

**PCR-based genotyping of rs3135388**. According to de Bakker et al. (2006)[12], the T allele at SNP locus rs3135388 is in almost complete LD with the HLA-DRB1*15:01 allele in the CEU population. With the designed SNP primers (Supplementary Table 1, Supplementary Fig. 1), the T/C variant rs3135388 was detected on the forward and reverse strand by Sanger sequencing. Three hundred and eight specimens (85 cases from St. Jørgen and 223 controls) were tested for this locus in the following manner: (1) For the St. Jørgen samples that provided HLA-DRB1 data (n=43), PCRs were performed independently at least two times per extract. (2) For the 26 St. Jørgen samples for which no HLA-DRB1 data were available but SNP amplicons could be generated, PCRs were performed between 7 and 11 times per specimen using a minimum of two independent extracts (exceptions: sample G417 only five replicates, sample G859 only six replicates). (3) For the 16 remaining St. Jørgen samples, neither HLA-DRB1 data nor SNP genotypes could be generated. (4) For the controls, PCRs were performed independently at least two times per extract. For 43% of control samples, PCRs were performed independently at least four times.

Only individuals with consistent results were included in the statistical analysis. However, allelic drop-out is a common problem when amplifying aDNA[41], and a heterozygote might erroneously be determined as a homozygote[43,44]. We therefore randomly selected 32 specimens (marked in Supplementary Data 1) that showed homozygous genotypes (either TT or CC) after the first round of PCR. For each of them, we performed between 7 and 11 PCRs as recommended[43,44] using a minimum of two independent extracts. For each individual, the homozygous state was confirmed in all replicates.

**PCR-based human mtDNA analysis**. MtDNA studies were performed with all 308 samples by sequencing parts of the hypervariable regions I (150 bp and 183 bp) and II (253 bp)[45]. For a random subset of 27 samples, new aDNA extracts were generated and used for a replication. In these cases, the haplogroup classification was confirmed. Haplogroups were assigned with Haplogrep 2.0 (http://haplogrep.uibk.ac.at/). Haplogroups showing a quality score below 100% were manually re-evaluated by consulting the established mtDNA phylogeny (www.phylotree.org).

**Statistical association analysis**. For the recruitment of medieval controls, the required sample size was calculated with the software G*Power, v3.1.9.2. For the available 138 alleles of our medieval LL samples and under the assumption of an odds ratio of 2.0[4,5], another 250 alleles in controls are required to obtain an a priori power of 75% at a significance level of 0.05. The comparisons of allele frequencies of DRB1*15:01/rs3135388 between different populations were performed with Fisher's exact test with the software R, v3.2.2[46].

**Shotgun HTS**. For each sample, two double-stranded DNA sequencing libraries were prepared according to an established, but slightly modified protocol for multiplex high-throughput sequencing[47].

First library: UDG-treated libraries were prepared in a 50-μL volume containing 20 μL of DNA extract, 1× NEB buffer 2 (New England Biolabs), 300 μM dNTPs (each), 0.005 mg/mL BSA, 1mM ATP, 20 U T4 Polynucleotide Kinase (Thermo Fisher Scientific) and 3 U USER enzyme (Uracil-Specific Excision Reagent, New England Biolabs). The reaction mix was incubated at 37 °C for 3 h. Subsequently, 6 U T4 DNA Polymerase (Thermo Fisher Scientific) were added and the reaction mix was incubated at 25 °C for 30 min and at 10 °C for 5 min.

Second library: non-UDG-treated libraries were prepared in a 50-μL volume containing 20 μL of DNA extract, 1× NEB buffer 2 (New England Biolabs), 300 μM dNTPs (each), 0.005 mg/mL BSA, 1mM ATP, 20 U T4 Polynucleotide Kinase

(Thermo Fisher Scientific) and 1.2 U T4 Polymerase (New England Biolabs). The mixture was incubated at room temperature for 30 min.

Both library preparations were continued as follows: After purification with the MinElute PCR Purification Kit (Qiagen) (elution volume 18 μL), adapter ligation was done in a 40-μL volume containing 18 μL DNA, 1× Quick Ligase buffer (New England Biolabs), 2.5 μM adapter mix (Solexa) and 0.5 U Quick Ligase (New England Biolabs). The mix was incubated at room temperature for 20 min. After another MinElute purification step (elution volume 20 μL), adapter fill-in was performed in a 40-μL volume containing 1× ThermoPol buffer (Thermo Fisher Scientific), 125 μM dNTPs and 16 U BSM DNA Polymerase (Thermo Fisher Scientific). The reaction mix was incubated at 37 °C for 20 min and then at 80 °C for 20 min.

Sample-specific indices were added to both library adapters via amplification with two index primers (i7, i5). Extraction and library blanks were treated in the same manner.

For UDG-treated libraries, indexing PCRs were performed in a 50-μL volume containing 10 μL template DNA, 1× AccuPrime Pfx reaction mix (Thermo Fisher Scientific), 1.25 U AccuPrime Pfx DNA Polymerase (Thermo Fisher Scientific), 0.3 μM P5 DNA primer and 0.3 μM P7 DNA primer. PCR conditions were as follows: 95 °C for 2 min, 10 cycles of (95 °C for 15 seconds, 60 °C for 30 seconds, 68 °C for 70 seconds).

For non-UDG-treated libraries, indexing PCRs were performed in a 50 μL volume containing 10 μL template DNA, 1× PfuTurbo Cx reaction buffer, 200 μM dNTPs, 2.5 U PfuTurbo Cx Hotstart DNA Polymerase (Agilent Technologies), 0.3 μM i7 DNA primer and 0.3 μM i5 DNA primer. PCR conditions were as follows: 95 °C for 2 min, 10 cycles of (95 °C for 30 seconds, 55 °C for 30 seconds, 72 °C for 60 seconds), elongation step of 72 °C for 10 min.

Both indexed DNA libraries were purified with the MinElute PCR Purification Kit (Qiagen) according to the manufacturer's instructions and eluted in 50 μL elution buffer.

A second amplification step was performed for all indexed libraries in 50-μL reactions containing 5 μL indexed library template, 1.25 U AccuPrime Pfx DNA Polymerase (Thermo Fisher Scientific), 1× AccuPrime Pfx reaction mix (Thermo Fisher Scientific) and 0.3 μM IS5 (5′-AATGATACGGCGACCACCGA-3′) and IS6 (5′-CAAGCAGAAGACGGCATACGA-3′) primers that bind to the adapters of the indexed libraries. Amplified products were purified with the MinElute PCR Purification Kit (Qiagen) (elution volume 53 μL) and quantified using the Agilent 2100 Bioanalyzer DNA 1000 chip. The sequencing was carried out on the Illumina HiSeq 2500 (2 × 125 bp) and HiSeq 4000 (2 × 75 bp) platform at the Institute of Clinical Molecular Biology, Kiel University, using the HiSeq v4 chemistry and the manufacturer's protocol for multiplex sequencing.

**HLA capture and HTS**. In this study, HLA regions were enriched with an in-solution bait capture and the SureSelectXT Target Enrichment System (Illumina) for the Illumina paired-end multiplexed sequencing library. Using a UDG-treated sequencing library and a custom bait library designed by Michael Wittig et al.[15], the classical class I (HLA-A, HLA-B, HLA-C) and class II HLA genes (HLA-DRB1, HLA-DQA1, HLA-DQB1, HLA-DPA, and HLA-DPB1) were enriched in 68 St. Jørgen samples. Samples were pooled (up to four) per capture with 800 ng of library DNA per pool in a volume of 3.4 μL. Hybridization buffer and blocking reagent were handled according to the manufacturer's instructions. As the captured samples were indexed already, 1 μL of each IS5 and IS6 primers (100 μM) were used for indexing/amplification instead of the primers provided in the capture kit. The post-capture PCR cycle number was set to 12. Purification of the amplified captured libraries was performed according to the protocol using AMPure XP beads. The quality of the captured library pools was assessed on the Agilent 2100 Bioanalyzer with the High Sensitivity DNA Assay. The sequencing was carried out on the Illumina HiSeq 4000 (2 × 75 cycles) platform at the Institute of Clinical Molecular Biology, Kiel University, using the HiSeq v4 chemistry and the manu-facturer's protocol for multiplex sequencing.

**Bioinformatic analysis**. Multiple HTS data sets were generated for the 68 individuals from St. Jørgen. The data sets were pre-processed (adapter clipping, merging, trimming) according to published protocols specific for aDNA using the EAGER pipeline[48].

**Genome-wide analysis of *M. leprae***. Multiple HTS data sets were generated for 68 of the 85 individuals from St. Jørgen. These data sets were pre-processed according to published protocols specific for aDNA[48]. The reads were then aligned to the *M. leprae* TN reference genomes applying established algorithms. Post-processing included the identification of genomic variation, effect prediction of SNPs, the de novo assembly of the bacterial genomes and metagenomic screening as well as phylogenetic analyses and the identification of HLA allele combinations. A detailed description of each step is given below.

Pre-processing of ancient genomes: The data sets produced for all ancient samples contained paired-end reads with varying numbers of overlapping nucleotides as well as artificial adapter sequences. We used ClipAndMerge v1.7.3, a module of the EAGER pipeline[48], to clip adapter sequences, merge corresponding paired-end reads in overlapping regions and to trim the resulting reads. These steps

are explained in detail below. We used the default options with the following command:

```
java -jar ClipAndMerge.jar -in1 $FASTQ1 -in2 $FASTQ2 \
-f AGATCGGAAGAGCACACGTCTGAACTCCAGTCAC \
-r AGATCGGAAGAGCGTCGTGTAGGGAAAGAGTGTA \
-l 25 -qt -q 20 -o $output_file
```

where $FASTQ1 and $FASTQ2 are the two gzipped FASTQ input files.

ClipAndMerge uses an overlap alignment of the respective forward or reverse adapter with the 3′ end of each read in order to remove sequencing adapter sequences. Regions at the 3′ end of each read that were contained in the alignment were clipped. Reads that were shorter than 25 nucleotides after adapter clipping or contained only adapter sequences (adapter dimmers) were removed. All remaining reads were then used in the merging step. Merging was performed for all remaining paired reads with a minimum overlap of 10 nucleotides and at most 5% mismatches in the overlap region. The algorithm selected the maximal overlap fulfilling these criteria. The consensus sequence was generated using the nucleotides in the overlap regions from the read with the higher PHRED quality score, maximizing the quality of the resulting read. In a final step, ClipAndMerge performed quality trimming of the reads and all nucleotides with PHRED scores smaller than 20 were trimmed from the 3′ end of each read. Finally, all reads with fewer than 25 nucleotides after quality trimming were removed. The resulting high-quality reads were used for the alignment against the *M. leprae* TN reference genome (NC_002677) with Bowtie2[49] v2.2.7. In this step, all reads were treated as single-end reads and mapping was performed using semi-global alignment mode and default parameters. The following command was used:

```
bowtie2 -t –mp 1,1 –ignore-quals –score-min L,0,-0.05 \
–no-unal -x $REF -U $FASTQ – S $SAM
```

where $REF is the reference FASTA file, $FASTQ is the gzipped input FASTQ file and $SAM is the output SAM file.

After the alignment, the SAM files were converted into BAM files, which were sorted and indexed using SAMtools v1.3 with default parameters and the following commands:

```
samtools view -h -q 0 -bS $SAM -o $BAM
samtools sort -o $OUT -T $TMP $BAM
samtools index $OUT
```

where $SAM is the bowtie2 output, $BAM is the converted BAM file, $TMP is a temporary file and $OUT is the final sorted BAM file.

All individual-specific final BAM files were concatenated using the MergeSamFiles algorithm of picard tools v1.139 (http://broadinstitute.github.io/picard) and indexed using SAMtools. We used the default parameters with the following commands:

```
java -jar picard.jar MergeSamFiles I=$BAMs O=$OUT
samtools index $OUT
```

where $BAMs is a string containing all individual-specific BAM files to be concatenated, separated by whitespace, and $OUT is the concatenated BAM file.

We used DeDup v0.9.9, part of the EAGER pipeline[48], to identify and remove all duplicate reads in the individual-specific BAM files with the default options and the following command:

```
java -jar DeDup.jar -i $IN -o $OUT
```

where $IN is the input BAM file and $OUT is the output BAM file.

To authenticate aDNA data sets, we evaluated the presence of postmortem DNA damage signatures from read alignments using mapDamage[50] v2.0.6 with default parameters and the following command:

```
mapDamage -v -i $BAM -r $REF -l 100 -d $PREFIX
```

where $BAM is the input BAM file containing only merged reads, $REF is the reference FASTA file used for the alignment and $PREFIX is a string containing the full path and an optional prefix for the output.

After alignment and duplicate removal, genomic variation was identified with the Genome Analysis Toolkit (GATK)[51] v3.6. First, a local realignment of the individual-specific BAM files was performed with the RealignerTargetCreator and the IndelRealigner modules of GATK[51]. Subsequently, the UnifiedGenotyper module was applied to call reference bases and variants from the alignment. Default parameters were used in the following commands:

```
java -jar GenomeAnalysisTK.jar -T RealignerTargetCreator \
-R $REF -I $BAM -o $INTERVALS
java -jar GenomeAnalysisTK.jar -T IndelRealigner \
-R $REF -I $BAM -targetIntervals $INTERVALS -o $REALIGNED
java -jar GenomeAnalysisTK.jar -T UnifiedGenotyper \
-R $REF -I $REALIGNED -o $VCF -mbq 15 -rf MappingQuality \
-mmq 20 -strand_call_conf 50 –sample_ploidy 2 -dcov 250 \
–output_mode EMIT_ALL_CONFIDENT_SITES
```

where the variable $REF is the FASTA file of the used reference genome, $BAM is the final alignment BAM file after duplicate removal, $INTERVALS are the target intervals for the local realignment, $REALIGNED is the output from the local realignment and $VFC is the output variant call set in VCF format.

To annotate an estimated effect of identified genomic variation, the software SnpEff[52] v4.2 was applied. We used the default parameters and the following command line:

```
java -jar snpEff.jar eff $GFF $VCF
```

where $GFF is a previously set up database based on a GFF file and $VCF is the out VCF file. We used the GFF file from NCBI for *M. leprae* (NC_002677) to build the database for SnpEff[52].

To analyze the effects of single SNPs, the VCF files containing the lists of annotated SNPs for each sample were merged using the following command available from VCFtools[53]:

```
vcf-merge sample1.vcf.gz [...] sampleX.vcf.gz | \
bgzip –c > merged.all.emit.confident.sites.vcf.gz
```

Variant annotations of upstream and downstream SNPs located further than 100 bases away from a gene were ignored. The annotations were manually completed with information about transcript type and function available on the NCBI and Mycobrowser databases. The results were compiled into a table containing information for each SNP regarding its effect on the genes in the strains in which the SNP occurs (Supplementary Data 2). In vivo phenotypic studies confirming the in silico annotations are rare because it is currently impossible to grow *M. leprae* in culture. Therefore, most of the predicted gene functions are based on comparisons with *M. tuberculosis* and numerous genes remain annotated as hypothetical proteins whose functions are unknown. Variants for which there is enough information available to perform in silico evaluation of the variant effects were selected by removing all variants with the following annotated effects: (1) Intergenic, upstream, downstream: The understanding of *M. leprae* intergenic regions is insufficient at present to estimate the effects of the intergenic variants on gene expression genes. (2) Hypothetical protein and pseudogene: No sufficient information was available on the transcripts. (3) Stable RNAs: The variants in annotated stable RNAs were removed so that the focus was on protein variants. (4) Variants present in only one genome: This filtering step was performed to focus on common variants.

Extracted mapping reads of ten high-coverage data sets (G34, G154, G404, G427, G507, G533, G722, G749, G1083, and G1149), covering the complete *M. leprae* reference genome (NC_002677.1) at least 10-fold, were de novo assembled using the SPAdes genome assembler[54] v3.5.0 with the following settings:

```
spades.py -t 8 -m 60 -k 121 –careful -s $IN -o $OUT
```

where $IN is a FASTQ file containing the mapped reads and $OUT is the output folder for SPAdes[54].

Other samples were not assembled as their respective data sets did not match the previously described threshold of a 10-fold coverage of the complete *M. leprae* reference genome. All possible values for the k parameter were tried with $k = 121$ yielding the best result with respect to the contig mean, N50, maximal contig size and number of produced contigs. The multiple genome alignment software Mauve v.2.4.0[55,56] was used to reorder the resulting contigs relative to the *M. leprae* TN reference genome for subsequent calculation of the genomic coverage. The reordering in Mauve[55,56] was executed with default parameters for the Mauve Contig Mover (MCM).

Shotgun sequencing data of multiple sequencing runs was pooled after pre-processing for the samples G34, G154, G404, G427, G507, G533, G722, G749, G1083, and G1149. The pooled reads were used to carry out a metagenomic de novo assembly with the Megahit assembler v1.0.3-8-g4b5271e[57]. The following command line was used:

```
megahit --presets meta -r IN -o OUT, where IN is a file containing the pre
```

−processed reads, OUT is the output directory.

For subsequent calculation of the genome coverage, contigs from the assemblies with a length of > 500 bases were blasted (v2.2.30) against the *M. leprae* strain TN genome. The following command line was used:

```
blastn -evalue 10e-6 -perc_identity 95 -outfmt "6 qseqid" \
-query $IN -out $OUT -db $REF
```

where $IN is a file containing the contigs, $OUT is the output file of blast and $REF is the blast database. Subsequently, the MCM was used as described above to calculate the genomic coverage for the contigs that mapped to the *M. leprae* TN genome.

**Metagenome screening**. All UDG-treated samples derived from teeth (*n*=68) were screened for their metagenomic content with the alignment tool MALT[14] and the metagenome analyzer MEGAN [58]. After pre-processing, as described above, MALT[14] v0_3_ln was used to align all samples against a collection of all complete bacterial genomes in FASTA format downloaded from the NCBI FTP server (ftp://ftp.ncbi.nlm.nih.gov/genomes/refseq/bacteria/). We used MALT[14] in BlastN mode with the following command line:

```
malt-run –inFile $IN –index $REF –output $OUT -id 85.0 -v \
-m BlastN -at SemiGlobal -top 1 -supp 0 \
-mq 100 -ssc -sps
```

where $IN is a FASTQ file after pre-processing, $REF is the MALT index, $OUT is output folder for MALT[14].

Subsequently, MEGAN[58] v6_4_15 was used to compute the taxonomical content, while pathogen screening was performed manually. Potentially interesting bacteria were selected for a specific alignment against the respective reference genome with Bowtie2[49] v2.2.7. The following parameters were used:

```
bowtie2 -t --mp 1,1 --ignore-quals --score-min L,0,-0.05 \
--no-unal --rg-id --rg SM:blank --rg PL:illumine
```

Genomic variation was identified in the genome-wide analysis of *M. leprae*. Variant positions (SNPs) as well as reference alleles were called in each data set if the quality was at least 40. The resulting VCF files were filtered to contain only

positions that were covered by at least five reads. Genome drafts were generated using the respective variant or reference alleles of these loci. Variant alleles were only used if the fraction of mapped reads containing the variation was at least 90%, otherwise the reference allele was used instead. If a reference locus was not sufficiently covered, the character "N" was inserted at the respective position.

A multiple genome alignment of 33 *M. leprae* genomes was computed using the progressive Mauve algorithm[55] integrated in the whole genome alignment software Mauve[56] v2.4.0. The sequences of the ten medieval genomes G34, G154, G404, G427, G507, G533, G722, G749, G1083, and G1149, five previously published medieval genomes (3077, Jorgen_625, Refshale_16, SK2, and SK8) and seventeen previously published modern genomes[2,21] (S2, S9, S10, S11, S13, S14, S15, Brw-01, Brw-05, Brw-10, Brw-12, Brw-20, Brw-25, TN, BR4923, Thai53, and NHDP63) were included in the alignment. Mauve's SNP export function was used to generate FASTA files for each data set that contained all sites that were variable among all genomes. These SNP assemblies were merged into a multi FASTA file and then used as input for MEGA7[59] to perform the phylogenetic analyses.

MEGA7[59] v7.0.18 was used to create maximum parsimony (MP), neighbor-joining (NJ) and maximum likelihood (ML) trees. All sequences included in the multiple alignment were used in all three tree reconstructions. In a separate analysis rooted MP, NJ, and ML trees were generated using *M. avium* 104 (NC_008595.1) as outgroup.

Close-Neighbor-Interchange algorithm (search level 1) was applied to construct the MP tree. Random addition of sequence (10 replicates) was used to obtain the initial trees. Maximum likelihood model selection analysis of the MEGA package (default parameters) was used to assess the best model for the evolutionary distances in the NJ and ML tree. The model with the lowest BIC value was chosen (Supplementary Data 3). The Tamura 3-parameter model[60] and uniform rate for all sites was the best model for the alignment without the outgroup. This model was also used for the alignment including the outgroup. The NJ and ML trees were then constructed using the Tamura 3-parameter model[60], uniform rates and default settings in MEGA[59]. Bootstrap values were inferred from 500 replicates in all three reconstruction methods.

The analysis involved 32 (33 with outgroup) nucleotide sequences. In the alignments, positions were not used that had less than 95% site coverage. The final data set comprised a total of 955 (547,078 with outgroup) informative positions.

**Genome-wide analysis of *Homo sapiens* and population genetics**. Each of the 68 individual-specific HTS data sets was pre-processed during the genome-wide analysis of *M. leprae*, as described above. After adapter clipping, merging, quality trimming, and quality control, the reads were additionally aligned to the human reference genome hg38. Post-processing included the identification of genomic variation, mtDNA haplotyping, sex assignment, principal component analysis and the estimation of IBD. A detailed description of these methods follows below.

After pre-processing, the resulting high-quality reads were mapped for each data set individually against the *H. sapiens* reference genome hg38 (GRCh38), downloaded from UCSC (http://hgdownload.soe.ucsc.edu/goldenPath/hg38/bigZips/) using Bowtie2[49] v2.2.7. In this step, all reads were treated as single-end reads and mapping was performed using semi-global alignment mode and default parameters. The following command was used:

bowtie2 -t –mp 1,1 –ignore-quals –score-min L,0,-0.05 \
–no-unal -x $REF -U $FASTQ – S $SAM

where $REF is the reference FASTA file, $FASTQ is the gzipped input FASTQ file and $SAM is the output SAM file.

After the alignment, the SAM files were converted into BAM files which were sorted and indexed using SAMtools v1.3 with default parameters and the following commands:

samtools view -h -q 0 -bS $SAM -o $BAM
samtools sort -o $OUT -T $TMP $BAM
samtools index $OUT

where $SAM is the bowtie2[49] output, $BAM is the converted BAM file, $TMP is a temporary file, and $OUT is the final sorted BAM file.

All individual-specific final BAM files were concatenated using the MergeSamFiles algorithm of picard tools v1.139 (http://broadinstitute.github.io/picard) and indexed with SAMtools. We used the default parameters with the following commands:

java -jar picard.jar MergeSamFiles I=$BAMs O=$OUT
samtools index $OUT

where $BAMs is a string containing all individual-specific BAM files to be concatenated, separated by whitespace, and $OUT is the concatenated BAM file.

We used DeDup v0.9.9, part of the EAGER pipeline[48], to identify and remove all duplicate reads in the individual-specific BAM files with the default options and the following command:

java -jar DeDup.jar -i $IN -o $OUT

where $IN is the input BAM file and $OUT is the output BAM file.

To authenticate aDNA data sets, we evaluated the presence of postmortem DNA damage signatures from read alignments using mapDamage[50] v2.0.6 with default parameters and the following command:

mapDamage -v -i $BAM -r $REF -l 100 -d $PREFIX

where $BAM is the input BAM file containing only merged reads, $REF is the reference FASTA file used for the alignment, and $PREFIX is a string containing the full path and an optional prefix for the output.

The alignment of high-quality reads against the *H. sapiens* reference genome hg38 was performed analogous to the *M. leprae* alignment using bowtie2[49] and the same settings as described above. The removal of duplicates and the evaluation of damage patterns with DeDup and mapDamage, respectively, were also performed according to the protocols described above. Genomic variation was identified using the haplotype caller module of GATK[51] v3.6 based on the alignment data sets. First, a local realignment of the individual-specific BAM files was performed with the RealignerTargetCreator and the IndelRealigner modules of GATK[51]. Subsequently, the UnifiedGenotyper module was applied to call reference bases and variants from the alignment. Default parameters were used in the following commands:

java -jar GenomeAnalysisTK.jar -T RealignerTargetCreator \
-R $REF -I $BAM -o $INTERVALS
java -jar GenomeAnalysisTK.jar -T IndelRealigner \
-R $REF -I $BAM -targetIntervals $INTERVALS -o $REALIGNED
java -jar GenomeAnalysisTK.jar -T UnifiedGenotyper \
-R $REF -I $REALIGNED -o $VCF -mbq 15 -rf MappingQuality \
-mmq 20 -strand_call_conf 50 –sample_ploidy 2 -dcov 250 \
–output_mode EMIT_ALL_CONFIDENT_SITES

where the variable $REF is the FASTA file of the used reference genome, $BAM is the final alignment BAM file after duplicate removal, $INTERVALS are the target intervals for the local realignment, $REALIGNED is the output from the local realignment, and $VFC is the output variant call set in VCF format.

For determination of sex, we computed the "read densities" dX, dY for the X and Y chromosomes, respectively, as the ratio of the number of reads mapped to the respective chromosome by its total chromosomal length (156,040,895 and 57,227,415 bp, respectively).

We applied principal component analysis (PCA) to pseudohomozygous calls obtained by randomly sampling alleles at genomic positions that are biallelic in the European individuals in the 1000 Genomes data set (release 20130502)[61]. Genotypes with alleles other than those observed in the European samples from the 1000 Genomes project were excluded as likely errors. We used the smartpca software (v13050) with default parameters for outlier detection (numoutlierevec 10, outliersigmathresh 6) to conduct the PCA. Overall, 463 of 503 European samples from the 1000 Genomes project and 53 of 68 individuals from St. Jørgen remained for analysis. The lsqproject parameter was used for projection of samples with differentially missing genotypes onto a single plot.

To assess the degree of relatedness between individuals, we used the software lcmlkin[62] v0.5.0 for estimating expected IBD sharing, $\hat{\pi}$, between pairs of individuals and the accompanying script SNPbam2vcf for computing genotype likelihoods. The set of SNPs included in these calculations was restricted to positions that were reported as being biallelic in the European 1000 Genomes populations. In addition, only those SNPs that were covered by at least four reads (after removal of duplicates) in at least two individuals were considered for the computations. Finally, default quality filters were applied using SNPbam2vcf (MQ $\geq$ 20, BQ $\geq$ 5, GQ $\geq$ 0.1).

**Identification of HLA allele combinations**. Even after enrichment and deep sequencing, the highly degraded and fragmented nature of aDNA led to a low and incomplete coverage of the HLA region. This prevents the use of most computational HLA genotyping algorithms that are designed for high-coverage and high-quality sequence data. Furthermore, the allele combinations present in the samples might include unknown alleles that are undetectable by approaches that compare the reads only to modern reference sequences. We therefore used a semi-manual approach, where we performed an automated read selection and sorting procedure, followed by manual filtering and allele identification. These steps are described in detail below.

To select all reads from DNA fragments belonging to the HLA class II region in our data sets, we first generated a comprehensive reference set, containing all known four-digit (2nd field resolution) alleles of the exon coding for the peptide-binding groove of the classical HLA class II genes (HLA-DPA1, -DPB1, -DQA1, -DQB1, -DRA, -DRB1). In order to prevent mis-mapping and the resulting mis-identification of reads due to paralogous sequence similarity, we also included all known corresponding alleles for the non-classical genes HLA-DRB3, -DRB4, -DRB5, -DRB6, -DRB7, and -DRB9. Sequences were downloaded from the IMGT/HLA database[63] (accessed 28 July 15). In addition, we included a corresponding exon sequence of HLA-DQB2 from the human reference genome (not represented in the IMGT/HLA database), again to prevent their reads to be mis-identified as HLA-DQB1 variants. We did not consider intron sequences, as these evolve rather neutrally and may contain ancient variation that could be misleading during the identification of functional HLA alleles. All sequence variants for a given locus were initially aligned using MUSCLE[64] and then evaluated manually in BioEdit[65].

The actual read alignment was performed using Bowtie2[49] v2.2.7 in local alignment and '-a reporting mode', allowing each read to map against multiple alleles. We used the following command:

bowtie2 -a -t --ma 1 --mp 1,1 --local --ignore-quals \
--score-min L,0,1 --no-unal -x $REF -U $FASTQ –S $SAM

where $REF is the reference FASTA file, $FASTQ is the gzipped input FASTQ file and $SAM is the output SAM file.

The resulting alignment contained all mapped reads and, in case of an unspecific mapping, multiple instances of the same read sequence, one for each

distinct mapping locus. All instances of the same sequence fragment (representing PCR duplicates) were noted and then collapsed per gene. The name and number of all mapping loci and alleles, respectively, were also noted. The sequences were then grouped by gene specificity—for each gene, the resulting set of sequences was inversely ordered according to the number of genes they would map to and then sorted by the starting position within the corresponding allele (Supplementary Fig. 7). This set of sequences was saved to a FASTA file for each sample.

Sample-specific FASTA files were used for each HLA gene to generate consensus sequences of the allele combination present in the sample. The GUI-based sequence alignment editor BioEdit v7.2.5[65] was used to display and sort the sequence reads representing the two alleles and identify sequences with PCR/sequencing errors (Supplementary Fig. 7). Overlapping reads that shared the same combination of variation were collapsed into a consensus sequence. If a read showed variants that were not supported by other overlapping reads, most likely representing PCR or sequencing errors, the read was discarded. Most emphasis was given to reads that were specific to the evaluated locus (i.e., did not map to other HLA genes) and to reads that were represented by multiple exact PCR duplicates (making it less likely that they represent sequencing errors).

The consensus sequences were manually compared with a reference alignment of all known four-digit alleles of the corresponding HLA gene and matching alleles were identified. Here we made use of the allele frequency information implicitly included in the HLA nomenclature: As four-digit alleles were named with a consecutive numbering scheme in the order of their discovery, the second set of digits roughly indicates their frequency (common alleles were discovered earlier than rare alleles). We therefore focused first on the first 10 four-digit alleles of each two-digit allele group, and if we identified one or more alleles that perfectly matched the given consensus sequences, we did not look for additional matches in the rarer alleles. In case of multiple equally well matching known alleles, we first recorded all of them, but eventually reported only the two-digit allele name (we never found equally well matching alleles from different two-digit allele groups). The identified full-length allele sequences were then again compared to a sample's given read alignment to verify that the identified alleles were supported by all high confidence reads. As there was an a priori expectation to find certain DRB1 alleles among the leprosy cases, we recoded the names of all known alleles in the DRB1 reference alignment to allow for an observer-blinded allele call. The true allele names were only revealed after allele calls for DRB1 had been completed.

**Replication of manual allele call for DRB1.** The manual read filtering and allele call for each sample was performed by one of three different researchers. In order to evaluate reproducibility of our allele call approach, we obtained consistent DRB1 genotype calls by two different researchers for 14 of the 68 samples. Of the 28 allele calls, 28 (100%) matched at the two-digit level (representing functional serotypes) and 14 (50%) matched exactly at the four-digit level (identical protein sequences). In 12 of the 14 called alleles that did not match exactly, the two replicate calls could not identify a unique four-digit allele sequence but overlapped in the range of possible four-digit alleles. Only in two cases did the allele calls lead to two different four-digit alleles, resulting in one and two nucleotide mismatches, respectively, over the length of the entire typed exon sequence (270 bp). Overall, this results in 99.96% reproducibility at the nucleotide level (3/7560 mismatches).

**Prediction of *M. leprae* peptide binding by HLA-DRB1 variants**. We used the IEDB-AR consensus algorithm[66] to computationally predict the repertoire of bound *M. leprae* peptides for all DRB1 variants (at 2nd field resolution) with a frequency of >1% in representative population samples from Schleswig-Holstein/Germany ($n = 129,336$) and a Danish minority population from northern Germany ($n = 918$), both included in the DKMS database. We first predicted DRB1 allele-specific binding (rank threshold for binding: ≤0.5) to all possible 15-mer peptides ($n = 5345$) of 19 *M. leprae* outer membrane proteins that were previously identified as likely antigenic, harboring known T- and B-cell epitopes[67] (Fig. 3). We subsequently also predicted DRB1 allele-specific binding to all possible 15-mer peptides using the entire *M. leprae* proteome ($n = 516,303$ peptides; Ensembl accession: ASM19585v1, Supplementary Fig. 9).

**Data availability**. Raw sequence read files have been deposited at the European Nucleotide Archive under accession no. ERP021830 (https://www.ebi.ac.uk/ena/data/view/PRJEB19769).

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

## Acknowledgements

We are grateful to the following people and institutions for providing samples, support, and advice: Johannes Krause, Alexander Herbig, Verena Schünemann, Horsens Museum, Museum Lolland-Falster, Odense Bys Museer, Sydvestjyske Museer, Archäologisches Landesmuseum der Stiftung Schleswig Holsteinische Landesmuseen Schloss Gottorf. This work was supported by the Graduate School Human Development in Landscapes, Cluster of Excellence Inflammation at Interfaces, and the Medical Faculty of Kiel University. We acknowledge financial support by Land Schleswig-Holstein within the funding programme Open Access Publikationsfonds. T.L.L. was supported by the German Research Foundation (DFG, grant LE2593/3-1).

## Author contributions

B.K.-K., J.L.B., T.L.L., and A.N. designed the experiment. B.K.-K., L.B., D.D.P., S.-C.K., M.B., L.M., J.S., P.T., E.B., M.W. performed the experiment. B.K.-K. M.Nu., L.B., M.B., D. D., J.L.B. M.No., A.C. F.P., T.L.L. analyzed the data. B.W., A.S., J.S. provided comparative data. A.F., S.S. provided research infrastructure. B.K.-K., J.L.B., D.D., M.No., A.C., T.L.L., and A.N. interpreted the results. B.K.-K, M.Nu., L.B., J.L.B., D.D., M.No., J.S., A.C., A.F., T.L.L., and A.N. wrote the manuscript. B.K.-K., T.L.L., and A.N. revised the paper.

## Additional information

**Competing interests:** The authors declare no competing interests.

