## [Peer Review File · Nature Communications]

Reviewer #1 (Remarks to the Author):

Krause-Kyora et al

Ancient DNA study reveals HLA susceptibility loci for leprosy in medieval Europeans

This manuscript presents analysis of association between the HLA-DRB1 15:01 allele and *M. leprae* infections in medieval Europe. Using DNA capture and snp-tagging the authors are able to call HLA alleles from 69 ancient human samples from the Skt Jørgen site with signs of lepromatous leprosy (LL) bone lesions and from 152 other non-LL medieval samples. The frequency of HLA-DRB1 15:01 in the LL positive samples is compared to the medieval controls as well as a contemporary Northern Germany population and found to be significantly higher in the LL positive cases. Additionally, ten of the *M. leprae* strains are assembled using three different approaches (alignment, reference binned de novo assembly and metagenomic de novo assembly) and the phylogenetic relationship to other ancient and modern strains established.

First of all I really like the study, I think the manuscript is written in a very clear way and the message comes out very convincing. However, I am concerned that I was not able to replicate the main association finding (see below).

The manuscript is very strict in terms of what is in the main text and what is in the supplement. If there is space to allow it, it would really great to see a bit of the methods/results from the supplement in the main text. The figures in the main text are single panes only and could easily be combined to multi-plot figures to allow for more results presented in figures.

The aDNA analysis looks very solid, the approach of both alignment to reference, de novo assembly of reference-binned reads and full metagenomic assembly showing 100% coverage of the reference genome (in case of the alignment) looks very convincing.

Major comments:

I am confused about the number of DRB1 alleles found in the dataset. In Table 1 it says: 39x DRB1 15:01 alleles out of 69 individuals (138 alleles in total) with a frequency of $69/138=0.283$. However, in both Table S1 and Table S20, that has the overview of the 69 ancient individuals from St Jørgen, there are only 13 DRB1*15:01 alleles? Additionally, when counting total alleles called (eg. not counting alleles with no call) at the DRB1 loci at St Jørgen (Table S20) there are a total of 82 alleles called, this gives a frequency of $13/82 = 0.158$. First, it looks like individuals with no allele assigned

were used as part of the total observed amount of alleles and second the actual frequency in the ancient St Jørgen LL data is close to modern Germany today (0.138)?

As I understand it there are 40 samples with positive *M. leprae* PCR hit (Table S1). Table S15 has *M. leprae* hits from the HTS-MALT analysis, but how can you be sure that all samples actually contain *M. leprae* as 20 samples have less than 100 *M. leprae* reads. How robust is that to contamination from other *Mycobacteria*? Also, DNA damage patterns can be very hard to assess with so few reads, it would be great to see the actual plots for some of the samples.

Minor comments:

I could not find which tests the authors are using to calculate significance.

Why use only outer membrane proteins for the analysis of antigen-binding from *M. leprae*? I understand that these proteins are known to be hypothesized to be antigenic, but it would be interesting to see the same plots for the entire proteome of *M. leprae*.

For the phylogeny using Maximum Likelihood sites with >5% gaps are removed. ML approaches are perfectly fine with using gaps so all information could be included (except all gap positions).

Figure S2. There are a lot of Krona plots in Figure S2. It would be useful to have the data summarized in a table of identified OTUs instead of the plots – it is really hard/impossible to compare across 50+ figures. Additionally, the plots are layered pie-charts making it even more impossible to compare across samples.

Supplement, pg 10. The UnifiedGenotyper was used for identification of genomic variation in the *M. leprae* samples and I noticed that the sample ploidy is set to 2. Were heterozygote genotype calls (if any) filtered for the SNP effect analyses?

Supplement, pg. 15. Was the other ancient genomes processed in the same way as the genomes sequenced in this study?

Supplement, pg 16. Please add the commands used instead of writing that you followed the GATK Best Practices, the practices have changes and may change again.

In supplementary reference tables, it would increase readability if the positive/important ancient samples were marked/grouped.

Reviewer #2 (Remarks to the Author):

Please see the attached file.

Reviewer #3 (Remarks to the Author):

The authors intended to test if the DRB1*15:01 allele could have influenced the susceptibility to lepromatous leprosy in medieval Europeans. Genetic data were obtained from bones of lepromatous patients buried at an ancient leprosarium and from controls obtained from medieval cemeteries and present-day donors. Analysis showed an association of DRB1*15:01 allele with predisposition to lepromatous leprosy in medieval Europeans compared to contemporary and medieval controls. They also evaluated the genome of strains of *M. leprae* detected in the bones of leprosy patients by high-throughput sequencing and constructed a phylogenetic tree using additional medieval and modern strains.

The authors brings original and relevant data, as the conclusion that past epidemics, such as leprosy, could have influenced the frequency of alleles associated with modern chronic diseases; however some points need to be reviewed and/or better explored to become adequate for publication. Major and minor appointments to be considered or answered by authors are presented below.

Major appointments

1. Assuming that the authors have produced sequence/genotyping data for the entire MRC/HLA locus, the major question that rises upon reading the manuscript is to why only data for DRB1 has been analyzed. This is particularly relevant given that there are numerous reports showing association between leprosy and several HLA variants, including classes I and III. Did the authors test other HLA loci for association? If not, what were the criteria for selecting DRB1 as the only target for analysis? Of note, the literature references presented to support DRB1*15:01 allele as a major candidate is somewhat incomplete.
2. Both in the abstract and the conclusion the authors make inference about a DRB1*15:01-DQB1*06:02 haplotype; it seems that molecular data for this haplotype has been generated (ref. table S19); however, no results are presented and/or discussed. Please clarify.
3. A large segment of the manuscript is dedicated to a study of *M. leprae* molecular epidemiology/phylogeny that is not mentioned in the title/abstract, thus resulting in an overall report that lacks uniformity and clarity of the main message to be conveyed. In short, the feel is of two almost independent papers in one. Perhaps it would be advantageous to concentrate on the *M. leprae* genomic data just as a necessary tool to prove infection and genomic stability, leaving complementary analysis – such as phylogeny – to a subsequent report. Please consider.
4. The authors insist in classifying the ancient leprosy individuals studied as “LL”, a clinical form of disease defined as such only in 1966 in a classic paper by Ridley & Jopling. Also, the authors use expressions such as “severe” to define disease status – all this without having access to clinical/epidemiological/microbiological data necessary for such inferences. How did the authors conclude for cases being of “severe” leprosy, compatible with the LL form as described by R&J? Why not BL, for example?
5. In the supplementary data the authors indicate excavating more skeletons than the ones apparently used; throughout the text, different numbers are presented as samples analyzed (79? 69? 68?). The same inconsistency is observed for controls. Could the authors please clarify?
6. It is not clear in the discussion and conclusions which ways the DRB1*15:01-DQB1*06:02 haplotype and the results obtained could influence the frequency of genetic factors associated with modern inflammatory diseases.
7. Please justify the choice of minimum 10x coverage for de novo assemblies once larger coverage is usually required for this kind of approach.
8. Considering the use of the Sanger sequencing technique, why rs3135388-T was used for sequenced instead of actual DRB1*15:01?

Minor appointments

1. Could the authors elucidate if the genomes were or not enriched? Information in the manuscript (line 77) and supplementary information (page 17) are contradictory.

2. For the sake of precision, please substitute continents by actual endemic countries (lines 58 and 59);
3. Please include the meaning of “IBD” abbreviation also in the manuscript;
4. Please review the use of the word ‘major’ as used in line 40;
5. If acceptable by the journal definitions, please consider including a ‘methods’ subtopic. Also, it could be helpful to have flow chart describing the experimental design included as supplementary information;
6. Please check if there is any more recent (than 1944) reference for leprosy diagnosis by bone analysis;
7. The word ‘loci’ needs to be in italic, please adjust. On the same note, please check if all the occurrences of *M. leprae* in the text are in italic;
8. It is not clear in the manuscript what was the biological material used to obtain the present-day controls. Please comment;
9. Line 165: please consider substituting the word ‘victims’ by ‘patients’ or similar;
10. Many references in the supplementary information do not correspond with the subject cited in the text (e.g. ‘arm positions’ (3) and ‘osteological analyses’ (16)); please perform a careful inspection of all references in the supplementary information.
11. Please include p values and OR in figure 4.
12. S1 table: a) sex determination by bones and molecular tests are divergent, please justify the choice; b) please provide a properly legend for sex, especially for ‘C’ and ‘?’; c) Instead of the green tag in TagSNP, we suggest adding a column containing the number of PCRs for each case; d) what is the meaning of ‘n’ and ‘T’ in the TagSNP column? e) why there are little information about age and sex in the ‘Revshade’ tab?
13. Could the authors please provide the following adjustments, as listed below?
 - Tables S12 and S14 have the same title;
 - Legend for ‘NA’ is missing in table S20;
 - In table S18 a ‘m’ is missing in the word ‘number’; also, please provide a legend for the colors and a description of the meaning of ‘pi hat’ and ‘nsnps’;
 - Table S7: legend for ‘nc’ is missing.
14. It would be interesting to include the commands for human sequencing analysis in the supplementary information, as done for the bacterial DNA analysis.

Point-by- point response to the referees

Editor comments:

We are particularly concerned about Reviewer #1's comments about conflicting numbers of DRB1 alleles found in the dataset and we would have to see this issue unambiguously resolved. We are further interested in Reviewer #3's suggestion of looking at other HLA variants (pt1).

We also note that your manuscript had initially been submitted in Letter format. We allow up to 5000 words (introduction, results, discussion) and up to ten main display items (i.e. figures and/or tables) in our manuscripts and we would like to invite you to make better use of this, which is also in line with a comment made by Reviewer #1.

Thank you for giving us the opportunity to submit an original research article. We have restructured and rewritten large parts of the manuscript.

Please note that in response to Reviewer #3's comment pt3 we would like to see a better synthesis of the two parts, rather than splitting into two separate papers.

We have followed Reviewer #3's suggestions and have adapted the manuscript accordingly (see our detailed reply to Reviewer #3).

Reviewer's comments:

Reviewer #1 (Remarks to the Author):

This manuscript present analysis of association between the HLA-DRB1 15:01 allele and *M. leprae* infections in medieval Europe. Using DNA capture and snp-tagging the authors are able to call HLA alleles from 69 ancient human samples from the Skt Jørgen site with signs of lepomatous leprosy (LL) bone lesions and from 152 other non-LL medieval samples. The frequency of HLA-DRB1 15:01 in the LL positive samples is compared to the medieval controls as well as a contemporary Northern Germany population and found to be significantly higher in the LL positive cases. Additionally, ten of the *M. leprae* strains are assembled using three different approaches (alignment, reference binned de novo assembly and metagenomic de novo assembly) and the phylogenetic relationship to other ancient and modern strains established.

First of all I really like the study, I think the manuscript is written in a very clear way and the message comes out very convincing. However, I am concerned that I was not able to replicate the main association finding (see below).

The manuscript is very strict in terms of what is in the main text and what is in the supplement. If there is space to allow it, it would really great to see a bit of the methods/results from the supplement in the main text. The figures in the main text are single panes only and could easily be combined to multi-plot figures to allow for more results presented in figures.

The aDNA analysis looks very solid, the approach of both alignment to reference, de novo assembly of reference-binned reads and full metagenomic assembly showing 100% coverage of the reference genome (in case of the alignment) looks very convincing.

Major comments:

I am confused about the number of DRB1 alleles found in the dataset. In Table 1 it says: 39xDRB1 15:01 alleles out of 69 individuals (138 alleles in total) with a frequency of $39/138=0.283$. However, in both Table S1 and Table S20, that has the overview of the 69 ancient individuals from St Jørgen, there are only 13 DRB1*15:01 alleles? Additionally, when counting total alleles called (eg. not counting alleles with no call) at the DRB1 loci at St Jørgen (Table S20) there are a total of 82 alleles called, this gives a frequency of $13/82 = 0.158$. First, it looks like individuals with no allele assigned were used as part of the total observed amount of alleles and second the actual frequency in the ancient St Jørgen LL data is close to modern Germany today (0.138)?

The aDNA association study was performed using SNP rs3135388 data obtained from medieval LL cases and contemporaneous controls. We generated genotype data for 69 cases (= 138 alleles) and 223 controls; the frequency for the rs3135388 allele T (n= 39 / 138 alleles) in cases amounted to 0.283 (Table 1). The T allele is used here as a reliable marker for genotyping DRB1*15:01.

In Table S1 and Table S20, we have listed the actual DRB1*15:01 calls in the cases determined after HLA capture and HTS. At the 4-digit level, we were able to classify 13 DRB1*15:01 alleles out of 57 DRB1 alleles (0.228). At the 2-digit level, we called 20 DRB1*15 alleles out of 82 (0.244). Most likely, all DRB1*15 alleles also represent DRB1*15:01 alleles (s. Results section, pages 6-7).

The observed frequencies of 0.228 or 0.244 are close to that measured by the SNP rs3135388 (0.283) and it is much higher than that found in Germans today (0.138). The discrepancy between the observed DRB1*15 / DRB1*15:01 frequencies and the rs3135388-T allele data is due to the much smaller sample sizes – that is, fewer LL cases were successful for the HLA sequencing than for SNP genotyping.

All this information is now laid out much more clearly in the rewritten manuscript and we hope that no further confusion will arise from this.

As I understand it there are 40 samples with positive M. leprae PCR hit (Table S1). Table S15 has M. leprae hits from the HTS-MALT analysis, but how can you be sure that all samples actually contain M. leprae as 20 samples have less than 100 M. leprae reads. How robust is that to contamination from other Mycobacteria? Also, DNA damage patterns can be very hard to assess with so few reads, it would be great to see the actual plots for some of the samples.

The definition of LL cases is based on several criteria:

1) Specific bone lesions, seen in all collected skeletons.

2) *M. leprae* DNA-positive in the screening PCR (n = 40).

3) Detection of *M. leprae* reads after HTS. All 68 skeletons show more than 1000 reads that specifically map to *M. leprae* (see Table S4). That all these reads represent authentic aDNA can be seen in Table S6, where the damage patterns (percentage damage for the first and last 5' bases of a read) are shown. Since the *M. leprae* genome is so different from other Mycobacteria genomes, it can be assumed with a high level of certainty that even a couple of mapped reads are diagnostic for leprosy.

We used different criteria (damage pattern of the reads, distribution of mapped reads over the *M. leprae* genome, reads mapping in unique regions of *M. leprae*) for the authentication of reads in the metagenomics data as recommended and published in "Mining Metagenomic Data Sets for Ancient DNA: Recommended Protocols for Authentication. Key, F., Posth, C., Krause, J., Herbig, A., Bos, K., Trends Genet. 33: 508-520 (2017)." As the *M. leprae* genome is quite different from other Mycobacteria, a few reads that fulfill the authentication criteria can identify *M. leprae*.

Minor comments:

I could not find which tests the authors are using to calculate significance.

For the association analysis, the test used (i.e. two-sided Fisher's Exact Test) is mentioned several times in the text (see pages 5-6).

Why use only outer membrane proteins for the analysis of antigen-binding from *M. leprae*? I understand that these proteins are known to be hypothesized to be antigenic, but it would be interesting to see the same plots for the entire proteome of *M. leprae*.

This is indeed an interesting point, which we have now followed. When predicting binding of the relevant DRB1 alleles to the entire *M. leprae* proteome, we find that the allele DRB1*15:01 is still among the alleles binding the smallest number of *M. leprae* peptides, but that there are several other alleles that also bind only few *M. leprae* peptides. The observation that the failure of DRB1*15:01 to bind many *M. leprae* peptides is more extreme when only considering likely antigenic outer membrane proteins is supportive of our interpretation that this allele confers susceptibility, particularly as these proteins have been characterized independently of the context of HLA-peptide presentation. We have now provided the requested information in Figure S9 and refer to it in the main text:

"HLA binding-prediction for the entire *M. leprae* proteome (516,303 unique peptides) still revealed limited relative presentation capacity for DRB1*15:01, but to a lesser extent (supplementary information, Fig. S9). The fact that peptide binding of DRB1*15:01 is relatively more limited when focusing on potential antigenic proteins suggests that it might be particularly ineffective in the context of antigen presentation. Limited antigen presentation could impair specific immunity against *M. leprae* infections

and thus confer susceptibility to its carriers, which is exactly what we found in the association analysis above.”

For the phylogeny using Maximum Likelihood sites with >5% gaps are removed. ML approaches are perfectly fine with using gaps so all information could be included (except all gap positions).

We applied the same experimental settings as in our former leprosy aDNA study (Schünemann et al. 2013, Science). The ML approach with “>5% gaps are removed” is giving the same phylogeny as using the ML approach with complete deletion. We used the “>5% gaps are removed” threshold to reduce the number of missing data and ambiguous bases which are common in aDNA data.

Figure S2. There are a lot of Krona plots in Figure S2. It would be useful to have the data summarized in a table of identified OTUs instead of the plots – it is really hard/impossible to compare across 50+ figures. Additionally, the plots are layered pie-charts making it even more impossible to compare across samples.

The Krona plots were removed. For easier comparison, we have added bar plots (Fig. S2) and a heat map (Fig. S3) displaying the bacteria composition of the 68 samples.

Supplement, pg 10. The UnifiedGenotyper was used for identification of genomic variation in the *M. leprae* samples and I noticed that the sample ploidy is set to 2. Were heterozygote genotype calls (if any) filtered for the SNP effect analyses?

In our analysis, no heterozygous genotypes were called.

Supplement, pg. 15. Was the other ancient genomes processed in the same way as the genomes sequenced in this study?

Yes, the comparative genomes published by Schuenemann et al. 2013 were processed in the same manner.

Supplement, pg 16. Please add the commands used instead of writing that you followed the GATK Best Practices, the practices have changes and may change again.

We have added the commands as suggested (page 17).

In supplementary reference tables, it would increase readability if the positive/important ancient samples were marked/grouped.

The samples are listed in Table S1 according to our internal lab numbers. To increase readability, we have now marked in red the lab numbers of those samples that provided SNP genotype data.

Reviewer #2 (Remarks to the Author):

General comments

This is a very interesting study and represents a great deal of work. The number of individuals examined is impressive. In addition, ten specimens were sufficiently well preserved that they were analyzed directly without enrichment. A phylogenetic analysis was performed, that included medieval genomes and modern samples that have been published elsewhere. The study investigates HLA loci, particularly DRB1*15.01, in relation to the presence of lepromatous leprosy. As the study is based in Europe where indigenous leprosy is extinct, the study is based on a phylogenetic analysis of archaeological human skeletal remains from northern European populations, including a medieval leprosarium. It is unclear whether a similar study carried out elsewhere would find the same relationship of HLA loci with leprosy.

We absolutely agree with the reviewer. Further aDNA research is needed to clarify this.

For example, in a Mexican study (ref. 8) the authors conclude that the HLA loci of interest are HLA-DRB1*08 (lower in leprosy patients) and HLA-DRB1*01 (higher in leprosy patients).

In our study, we first and foremost focused on DRB1*15:01 as it is – within the HLA region – the strongest known leprosy risk factor in very diverse populations. Because of power considerations, we selected this locus as we had only a very restricted number of skeletons (affected and unaffected) at our disposal.

The HLA association study is an interesting development in the field of aDNA. In the cited references there appear to be differences between HLA loci and modern leprosy such as China and northern India versus Mexico. Of course, Mexico is of special interest due to the presence of *Mycobacterium lepromatosis*, although we now realize this has a wider distribution, including in modern red squirrels in the UK. The strains of *M. leprae* in modern British squirrels reveals that these are distinct from human leprosy but the strains may have diverged quite recently. Considering the historical leprosy cases, it is interesting that the authors identified a genotype 3K strain in Denmark as until now this genotype has only been found in central and southeast Europe. In history, wealthy individuals with leprosy are known to have travelled and gone on pilgrimages, so this may possibly explain why individual G507 acquired a different and distinct strain of *M. leprae*. No genome of a 3K historical strain of the *M. leprae* genotype has yet been published, but clearly this is an area where more data are required.

Yes, G507 carried an *M. leprae* strain that represents a 3K SNP-type / branch 0 member. This is indeed the first medieval complete genome of this type. We agree that more modern and ancient genomes are needed to resolve much better the current phylogeographic pattern.

Although the final assertion in the abstract is that ‘past epidemics such as leprosy influenced the frequency of alleles associated with chronic diseases today’, it is unclear how this may have

come about. Patients rarely die from leprosy although they may be more susceptible to other more dangerous infections, including tuberculosis. In the absence of appropriate antimicrobial therapy, due to nerve damage leprosy patients suffer from pathological changes including loss of sensation, blindness and inability to use their hands, for example. In the results and discussion the authors are more moderate and suggest that limited HLA-presentation of antigens may have impaired an *M. leprae*-specific response.

Thank you very much for pointing out his question. We have now included a short paragraph in the Discussion section (p. 10) where we discuss this issue of how leprosy might have acted as a potent selection pressure.

Typographical and grammatical comments

Reference 11: Herbig et al. This is a preprint published in 2016.

This article is available on bioRxiv and is citable.

Supplementary references:

57: This has now been published in the journal *Fly* in 2012.

We have included the article in the reference list of the supplements.

72: Schäfer et al. Instead of citing a website that requires a password, it is suggested that the 2017 reference is given as it has now been published in *BMC Bioinformatics* (2017).

We have included the article in the reference list of the supplements.

Reviewer #3 (Remarks to the Author):

The authors intended to test if the DRB1*15:01 allele could have influenced the susceptibility to lepromatous leprosy in medieval Europeans. Genetic data were obtained from bones of lepromatous patients buried at an ancient leprosarium and from controls obtained from medieval cemeteries and present-day donors. Analysis showed an association of DRB1*15:01 allele with predisposition to lepromatous leprosy in medieval Europeans compared to contemporary and medieval controls. They also evaluated the genome of strains of *M. leprae* detected in the bones of leprosy patients by high-throughput sequencing and constructed a phylogenetic tree using additional medieval and modern strains.

The authors brings original and relevant data, as the conclusion that past epidemics, such as leprosy, could have influenced the frequency of alleles associated with modern chronic diseases; however some points need to be reviewed and/or better explored to become adequate for publication. Major and minor appointments to be considered or answered by authors are presented below.

Major appointments

1. Assuming that the authors have produced sequence/genotyping data for the entire MRC/HLA locus, the major question that rises upon reading the manuscript is to why only data for DRB1 has been analyzed. This is particularly relevant given that there are numerous reports showing association between leprosy and several HLA variants, including classes I and III. Did the authors test other HLA loci for association? If not, what were the criteria for selecting DRB1 as the only target for analysis? Of note, the literature references presented to support DRB1*15:01 allele as a major candidate is somewhat incomplete.

In our study, we focused on DRB1*15:01 as it is – within the HLA region – the strongest known leprosy risk factor in very diverse populations. We have described this more clearly in the manuscript now. In addition, we have targeted DRB1*15:01 as the SNP rs3135388 is known to be a reliable marker for DRB1*15:01. Because of power considerations and to avoid multiple testing, we selected only this locus as we had a very restricted number of skeletons (affected and unaffected) and too few ancient HLA genotypes for a sound statistical analysis. Thus, we did not test other loci for association with LL.

We have now cited 5 publications (Ref. 5-9) that refer to the association of DRB1*15:01 and leprosy. Moreover, the article by Jarduli et al. 2013 is a review paper that cites the most important case-control studies focusing on the HLA region and leprosy.

2. Both in the abstract and the conclusion the authors make inference about a DRB1*15:01-DQB1*06:02 haplotype; it seems that molecular data for this haplotype has been generated (ref. table S19); however, no results are presented and/or discussed. Please clarify.

We agree that this data is relevant and thank the reviewer for pointing this out. We have now included this information in the generally expanded results section on HLA allele calling:

“Interestingly, in all 13 cases where DRB1*15:01 could be called (representing 12 individuals, including one homozygote), we also found the allele DQB1*06:02, suggesting strong linkage disequilibrium between these alleles at the two loci (supplementary information, Table S20). These two alleles indeed define a DRB1-DQB1 haplotype that is still found in modern Europeans at considerable frequencies.”

3. A large segment of the manuscript is dedicated to a study of *M. leprae* molecular epidemiology/phylogeny that is not mentioned in the title/abstract, thus resulting in an overall report that lacks uniformity and clarity of the main message to be conveyed. In short, the feel is of two almost independent papers in one. Perhaps it would be advantageous to concentrate on the *M. leprae* genomic data just as a necessary tool to prove infection and genomic stability, leaving complementary analysis – such as phylogeny – to a subsequent report. Please consider.

As recommended by the reviewer, we have restructured the manuscript. The emphasis is now on the ancient HLA association study and the HLA profiling. The *M. leprae* genomic data is only

used to define the *M. leprae* DNA-positive individuals as cases, to point out the stability of the *M. leprae* genomes over the last 1000 years and to assess aDNA damage patterns. The phylogeography and phylogeny of the observed *M. leprae* strains are not discussed – this information will be included in another manuscript.

4. The authors insist in classifying the ancient leprosy individuals studied as “LL”, a clinical form of disease defined as such only in 1966 in a classic paper by Ridley & Jopling. Also, the authors use expressions such as “severe” to define disease status – all this without having access to clinical/epidemiological/microbiological data necessary for such inferences. How did the authors conclude for cases being of “severe” leprosy, compatible with the LL form as described by R&J? Why not BL, for example?

According to Ridley & Jopling (1966) only the severe form of leprosy (LL) leads to bone lesions. This is supported by numerous other studies (e.g. Andersen et al. 1994, *Int J Osteoarchaeology* 4: 21-30; Lastoria & Milanez Morgado de Abreu 2014, *An Bras Dermatol* 89: 205-218). Specific bone lesions are routinely used as diagnostic criteria for the classification of LL in archaeo-anthropological material (e.g. Inskip et al. 2015, *PLoS One* 10: e0124282; Andersen et al. 1994). The classification of our cases as LL is described in detail by Boldsen 2001 (*J Phys Anthropol* 115: 380-387).

5. In the supplementary data the authors indicate excavating more skeletons than the ones apparently used; throughout the text, different numbers are presented as samples analyzed (79? 69? 68?). The same inconsistency is observed for controls. Could the authors please clarify?

We did not mention the total number of excavated individuals per site. The number of samples that were collected for analysis in our study are mentioned in the Supplementary Information (page 3-4) and in the Table S1 (St Jørgen n=85; Ribe n=42; Revshale n=45; Tirup n=57; Rathausmarkt n=79).

In the manuscript, we have re-written the Results section to facilitate the readability of the text and to highlight which sample sizes were used for which analysis (e.g. rs3135388 data for the association study, DRB1*15:01 data in the HLA profiling).

6. It is not clear in the discussion and conclusions which ways the DRB1*15:01-DQB1*06:02 haplotype and the results obtained could influence the frequency of genetic factors associated with modern inflammatory diseases.

We have now addressed this point more clearly in the discussion, outlining that leprosy should have led to a frequency decline of this allele and haplotype. We continue further that since it is still very common in modern populations, likely other fitness effects are also associated with this allele, counteracting the deleterious effect of leprosy in medieval times.

7. Please justify the choice of minimum 10x coverage for de novo assemblies once larger coverage is usually required for this kind of approach.

10 x is the threshold used in our former aDNA study (Schünemann et al. 2013, Science) and was shown to be sufficient. A 10 x coverage already allows researchers to reliably call variants in bacterial genomes. In ancient genome studies, it is quite exceptional to obtain such a high coverage from shotgun data.

8. Considering the use of the Sanger sequencing technique, why rs3135388-T was used for sequenced instead of actual DRB1*15:01?

The allele nomenclature for classical HLA alleles (e.g. DRB1*15:01) refers to unique protein sequences of the domains that shape the antigen-binding groove of the HLA molecule. In case of the DRB1 molecule, the binding groove is shaped by exon 2 of the DRB1 gene, which spans 267 bp to code for 89 amino acids. A target length of 267 bp (plus necessary flanking sequences for primer annealing) is far beyond the average DNA fragment size of ancient samples. And the exon itself is far too variable to design reliable primers inside the coding sequence. Direct sequencing of the exon sequences for HLA genes is thus only possible from modern high-quality DNA that is little fragmented.

Even modern SNP-based GWAS approaches have no SNP marker inside these variable exons, because the variability would lead to unreliable hybridization success. Classical HLA genotypes are then inferred by imputation, which is exactly what we are doing here (inferring presence of HLA allele from presence of nearby linked SNP allele). Note, however, that we are also verifying the linkage between this SNP allele and DRB1*15:01 in a subset of individuals for which both genotyping assays were successful.

We have now included a brief explanation of this point in the manuscript:

“Direct Sanger sequencing of the relevant DRB1 exon is not possible from aDNA, because the length of the exon far exceeds the average DNA fragment size of ancient samples and the polymorphic exon sequence precludes the use of intra-exon primers.”

Minor appointments

1. Could the authors elucidate if the genomes were or not enriched? Information in the manuscript (line 77) and supplementary information (page 17) are contradictory.

The *M. leprae* and human genomes were generated after shotgun HTS without enrichment (p. 6, paragraph *M. leprae* genome analysis and metagenomic screening). The HLA region was enriched by capture before HTS (p. 6, paragraph *HTS-based HLA typing*).

2. For the sake of precision, please substitute continents by actual endemic countries (lines 58 and 59);

We have included some countries as example for each geographical region with the highest total number of new cases per year (page 3).

3. Please include the meaning of “IBD” abbreviation also in the manuscript;

We have explained the abbreviation IBD (identical by descent) in the manuscript (page 10).

4. Please review the use of the word ‘major’ as used in line 40;

It has been replaced by the word strong.

5. If acceptable by the journal definitions, please consider including a ‘methods’ subtopic. Also, it could be helpful to have flow chart describing the experimental design included as supplementary information;

We have rewritten the paper to increase the readability of the manuscript and included a well-structured methods section that describes the workflow of the experiments (p. 12-17).

6. Please check if there is any more recent (than 1944) reference for leprosy diagnosis by bone analysis;

We have replaced the 1944 reference by a more recent publication (Ref 1 and 10).

7. The word ‘loci’ needs to be in italic, please adjust. On the same note, please check if all the occurrences of *M. leprae* in the text are in italic;

Thank you - we have corrected the writing of Latin words throughout the manuscript.

8. It is not clear in the manuscript what was the biological material used to obtain the present-day controls. Please comment;

The present-day data are provided by the DKMS, a non-profit organization, recruiting potential volunteer donors for hematopoietic stem cell transplantation. Donors have been routinely typed at high resolution for HLA-A, -B, -C, -DRB1, -DQB1, and -DPB1 upon registration. DKMS also records self-assessed parentage, categorized by country of origin. Mainly saliva sample are used as biological material (details see Supplementary information, page 21).

9. Line 165: please consider substituting the word ‘victims’ by ‘patients’ or similar;

We have changed the wording as suggested.

10. Many references in the supplementary information do not correspond with the subject cited in the text (e.g. ‘arm positions’ (3) and ‘osteological analyses’ (16)); please perform a careful inspection of all references in the supplementary information.

Thank you, unfortunately, during reformatting the order of references was shifted. We have now carefully reviewed and corrected the reference list.

11. Please include p values and OR in figure 4.

We have included p values and OR in Figure 2 (former Figure 4).

12. S1 table: a) sex determination by bones and molecular tests are divergent, please justify the choice; b) please provide a properly legend for sex, especially for 'C' and '?'; c) Instead of the green tag in TagSNP, we suggest adding a column containing the number of PCRs for each case; d) what is the meaning of 'n' and 'T' in the TagSNP column? e) why there are little information about age and sex in the 'Revshade' tab?

As suggested by the reviewer, we have made the following changes to Table S1:

a) We have reported the results of both methods in order to be transparent. Inconsistent results in sex determination (osteology vs. molecular typing) are commonly observed. The sex determination based on osteological measurements can be imprecise, especially if the material is not completely preserved.

b) A proper legend is now provided (? = osteological sex determination was not possible).

c) We have added a column with the number of PCRs for each case.

d) We have provided a proper legend (n= no data available, T = T allele)

e) Only the samples that were positive for the genetic analysis were analyzed by the anthropologist.

13. Could the authors please provide the following adjustments, as listed below?

- Tables S12 and S14 have the same title;

- Legend for 'NA' is missing in table S20;

- In table S18 a 'm' is missing in the word 'number'; also, please provide a legend for the colors and a description of the meaning of 'pi hat' and 'nsnps';

- Table S7: legend for 'nc' is missing.

We thank the reviewer for pointing out these mistakes; they have been corrected in the supplementary.

14. It would be interesting to include the commands for human sequencing analysis in the supplementary information, as done for the bacterial DNA analysis.

The commands for the human sequencing analysis have been included in the supplementary (pages 16 and 17).

Point-by-point response to the referees

Editor comments:

We are particularly concerned about Reviewer #1's comments about conflicting numbers of DRB1 alleles found in the dataset and we would have to see this issue unambiguously resolved. We are further interested in Reviewer #3's suggestion of looking at other HLA variants (pt1).

We also note that your manuscript had initially been submitted in Letter format. We allow up to 5000 words (introduction, results, discussion) and up to ten main display items (i.e. figures and/or tables) in our manuscripts and we would like to invite you to make better use of this, which is also in line with a comment made by Reviewer #1.

Thank you for giving us the opportunity to submit an original research article. As we have restructured and rewritten large parts of the manuscript, we have not used track changes. Since this would be more confusing than helpful.

Please note that in response to Reviewer #3's comment pt3 we would like to see a better synthesis of the two parts, rather than splitting into two separate papers.

We have followed the Editor's suggestions and have synthesized better the two parts (*M. leprae* genomics and HLA association study) of the manuscript (see our detailed reply to Reviewer #3).

Reviewer's comments:

Reviewer #1 (Remarks to the Author):

This manuscript present analysis of association between the HLA-DRB1 15:01 allele and *M. leprae* infections in medieval Europe. Using DNA capture and snp-tagging the authors are able to call HLA alleles from 69 ancient human samples from the Skt Jørgen site with signs of lepomatous leprosy (LL) bone lesions and from 152 other non-LL medieval samples. The frequency of HLA-DRB1 15:01 in the LL positive samples is compared to the medieval controls as well as a contemporary Northern Germany population and found to be significantly higher in the LL positive cases. Additionally, ten of the *M. leprae* strains are assembled using three different approaches (alignment, reference binned de novo assembly and metagenomic de novo assembly) and the phylogenetic relationship to other ancient and modern strains established.

First of all I really like the study, I think the manuscript is written in a very clear way and the message comes out very convincing. However, I am concerned that I was not able to replicate the main association finding (see below).

The manuscript is very strict in terms of what is in the main text and what is in the supplement. If there is space to allow it, it would really great to see a bit of the methods/results from the

supplement in the main text. The figures in the main text are single panes only and could easily be combined to multi-plot figures to allow for more results presented in figures.

The aDNA analysis looks very solid, the approach of both alignment to reference, de novo assembly of reference-binned reads and full metagenomic assembly showing 100% coverage of the reference genome (in case of the alignment) looks very convincing.

Major comments:

I am confused about the number of DRB1 alleles found in the dataset. In Table 1 it says: 39xDRB1 15:01 alleles out of 69 individuals (138 alleles in total) with a frequency of $69/138=0.283$. However, in both Table S1 and Table S20, that has the overview of the 69 ancient individuals from St Jørgen, there are only 13 DRB1*15:01 alleles? Additionally, when counting total alleles called (eg. not counting alleles with no call) at the DRB1 loci at St Jørgen (Table S20) there are a total of 82 alleles called, this gives a frequency of $13/82 = 0.158$. First, it looks like individuals with no allele assigned were used as part of the total observed amount of alleles and second the actual frequency in the ancient St Jørgen LL data is close to modern Germany today (0.138)?

The aDNA association study was performed using SNP rs3135388 data obtained from medieval LL cases and contemporaneous controls. We generated genotype data for 69 cases (= 138 alleles) and 223 controls; the frequency for the rs3135388 allele T ($n= 39 / 138$ alleles) in cases amounted to 0.283 (Table 1). The T allele is used here as a reliable marker for genotyping DRB1*15:01.

In Table S1 and Table S20, we have listed the actual DRB1*15:01 calls in the cases determined after HLA capture and HTS. At the 4-digit level, we were able to classify 13 DRB1*15:01 alleles out of 57 DRB1 alleles (0.228). At the 2-digit level, we called 20 DRB1*15 alleles out of 82 (0.244). Most likely, all DRB1*15 alleles also represent DRB1*15:01 alleles (s. Results section, pages 5-8).

The observed frequencies of 0.228 or 0.244 are close to that measured by the SNP rs3135388 (0.283) and it is much higher than that found in Germans today (0.138). The discrepancy between the observed DRB1*15 / DRB1*15:01 frequencies and the rs3135388-T allele data is due to the much smaller sample sizes – that is, fewer LL cases were successful for the HLA sequencing than for SNP genotyping.

All this information is now laid out much more clearly in the rewritten manuscript and we hope that no further confusion will arise from this.

As I understand it there are 40 samples with positive M. leprae PCR hit (Table S1). Table S15 has M. leprae hits from the HTS-MALT analysis, but how can you be sure that all samples actually contain M. leprae as 20 samples have less than 100 M. leprae reads. How robust is that to contamination from other Mycobacteria? Also, DNA damage patterns can be very hard to assess with so few reads, it would be great to see the actual plots for some of the samples.

The definition of LL cases is based on several criteria:

- 1) Specific bone lesions, seen in all collected skeletons.
- 2) *M. leprae* DNA-positive in the screening PCR (n = 40).
- 3) Detection of *M. leprae* reads after HTS. All 68 skeletons show more than 1000 reads that specifically map to *M. leprae* (see Table S4). That all these reads represent authentic aDNA can be seen in Table S6, where the damage patterns (percentage damage for the first and last 5' bases of a read) are shown. Since the *M. leprae* genome is so different from other Mycobacteria genomes, it can be assumed with a high level of certainty that even a couple of mapped reads are diagnostic for leprosy.

We used different criteria (damage pattern of the reads, distribution of mapped reads over the *M. leprae* genome, reads mapping in unique regions of *M. leprae*) for the authentication of reads in the metagenomics data as recommended and published in "Mining Metagenomic Data Sets for Ancient DNA: Recommended Protocols for Authentication. Key, F., Posth, C., Krause, J., Herbig, A., Bos, K., Trends Genet. 33: 508-520 (2017)." As the *M. leprae* genome is quite different from other Mycobacteria, a few reads that fulfill the authentication criteria can identify *M. leprae*.

Minor comments:

I could not find which tests the authors are using to calculate significance.

For the association analysis, the test used (i.e. two-sided Fisher's Exact Test) is mentioned several times in the text (see pages 5-6).

Why use only outer membrane proteins for the analysis of antigen-binding from *M. leprae*? I understand that these proteins are known to be hypothesized to be antigenic, but it would be interesting to see the same plots for the entire proteome of *M. leprae*.

This is indeed an interesting point, which we have now followed. When predicting binding of the relevant DRB1 alleles to the entire *M. leprae* proteome, we find that the allele DRB1*15:01 is still among the alleles binding the smallest number of *M. leprae* peptides, but that there are several other alleles that also bind only few *M. leprae* peptides. The observation that the failure of DRB1*15:01 to bind many *M. leprae* peptides is more extreme when only considering likely antigenic outer membrane proteins is supportive of our interpretation that this allele confers susceptibility, particularly as these proteins have been characterized independently of the context of HLA-peptide presentation. We have now provided the requested information in Figure S9 and refer to it in the main text:

"HLA binding-prediction for the entire *M. leprae* proteome (516,303 unique peptides) still revealed limited relative presentation capacity for DRB1*15:01, but to a lesser extent (supplementary information, Fig. S9). The fact that peptide binding of DRB1*15:01 is relatively more limited when focusing on potential antigenic proteins suggests that it might be particularly ineffective in the context of antigen presentation.

Limited antigen presentation could impair specific immunity against *M. leprae* infections and thus confer susceptibility to its carriers, which is exactly what we found in the association analysis above.”

For the phylogeny using Maximum Likelihood sites with >5% gaps are removed. ML approaches are perfectly fine with using gaps so all information could be included (except all gap positions).

We applied the same experimental settings as in our former leprosy aDNA study (Schünemann et al. 2013, Science). The ML approach with “>5% gaps are removed” is giving the same phylogeny as using the ML approach with complete deletion. We used the “>5% gaps are removed” threshold to reduce the number of missing data and ambiguous bases which are common in aDNA data.

Figure S2. There are a lot of Krona plots in Figure S2. It would be useful to have the data summarized in a table of identified OTUs instead of the plots – it is really hard/impossible to compare across 50+ figures. Additionally, the plots are layered pie-charts making it even more impossible to compare across samples.

The Krona plots were removed. For easier comparison, we have added bar plots (Fig. S2) and a heat map (Fig. S3) displaying the bacteria composition of the 68 samples.

Supplement, pg 10. The UnifiedGenotyper was used for identification of genomic variation in the *M. leprae* samples and I noticed that the sample ploidy is set to 2. Were heterozygote genotype calls (if any) filtered for the SNP effect analyses?

In our analysis, no heterozygous genotypes were called.

Supplement, pg. 15. Was the other ancient genomes processed in the same way as the genomes sequenced in this study?

Yes, the comparative genomes published by Schuenemann et al. 2013 were processed in the same manner.

Supplement, pg 16. Please add the commands used instead of writing that you followed the GATK Best Practices, the practices have changes and may change again.

We have added the commands as suggested (page 17).

In supplementary reference tables, it would increase readability if the positive/important ancient samples were marked/grouped.

The samples are listed in Table S1 according to our internal lab numbers. To increase readability, we have now marked in red the lab numbers of those samples that provided SNP genotype data.

Reviewer #2 (Remarks to the Author):

General comments

This is a very interesting study and represents a great deal of work. The number of individuals examined is impressive. In addition, ten specimens were sufficiently well preserved that they were analyzed directly without enrichment. A phylogenetic analysis was performed, that included medieval genomes and modern samples that have been published elsewhere. The study investigates HLA loci, particularly DRB1*15.01, in relation to the presence of lepromatous leprosy. As the study is based in Europe where indigenous leprosy is extinct, the study is based on a phylogenetic analysis of archaeological human skeletal remains from northern European populations, including a medieval leprosarium. It is unclear whether a similar study carried out elsewhere would find the same relationship of HLA loci with leprosy.

We absolutely agree with the reviewer. Further aDNA research is needed to clarify this.

For example, in a Mexican study (ref. 8) the authors conclude that the HLA loci of interest are HLA-DRB1*08 (lower in leprosy patients) and HLA-DRB1*01 (higher in leprosy patients).

In our study, we first and foremost focused on DRB1*15:01 as it is – within the HLA region – the strongest known leprosy risk factor in very diverse populations. Because of power considerations, we selected this locus as we had only a very restricted number of skeletons (affected and unaffected) at our disposal.

The HLA association study is an interesting development in the field of aDNA. In the cited references there appear to be differences between HLA loci and modern leprosy such as China and northern India versus Mexico. Of course, Mexico is of special interest due to the presence of *Mycobacterium lepromatosis*, although we now realize this has a wider distribution, including in modern red squirrels in the UK. The strains of *M. leprae* in modern British squirrels reveals that these are distinct from human leprosy but the strains may have diverged quite recently. Considering the historical leprosy cases, it is interesting that the authors identified a genotype 3K strain in Denmark as until now this genotype has only been found in central and southeast Europe. In history, wealthy individuals with leprosy are known to have travelled and gone on pilgrimages, so this may possibly explain why individual G507 acquired a different and distinct strain of *M. leprae*. No genome of a 3K historical strain of the *M. leprae* genotype has yet been published, but clearly this is an area where more data are required.

Yes, G507 carried an *M. leprae* strain that represents a 3K SNP-type / branch 0 member. This is indeed the first medieval complete genome of this type. We agree that more modern and ancient genomes are needed to resolve much better the current phylogeographic pattern.

Although the final assertion in the abstract is that ‘past epidemics such as leprosy influenced the frequency of alleles associated with chronic diseases today’, it is unclear how this may have come about. Patients rarely die from leprosy although they may be more susceptible to other more dangerous infections, including tuberculosis. In the absence of appropriate antimicrobial therapy, due to nerve damage leprosy patients suffer from pathological changes including loss of sensation, blindness and inability to use their hands, for example. In the results and discussion the authors are more moderate and suggest that limited HLA-presentation of antigens may have impaired an *M. leprae*-specific response.

Thank you very much for pointing out his question. We have now included a short paragraph in the Discussion section (p. 10) where we discuss this issue of how leprosy might have acted as a potent selection pressure.

Typographical and grammatical comments

Reference 11: Herbig et al. This is a preprint published in 2016.

This article is available on bioRxiv and is citable.

Supplementary references:

57: This has now been published in the journal *Fly* in 2012.

We have included the article in the reference list of the supplements.

72: Schäfer et al. Instead of citing a website that requires a password, it is suggested that the 2017 reference is given as it has now been published in *BMC Bioinformatics* (2017).

We have included the article in the reference list of the supplements.

Reviewer #3 (Remarks to the Author):

The authors intended to test if the DRB1*15:01 allele could have influenced the susceptibility to lepromatous leprosy in medieval Europeans. Genetic data were obtained from bones of lepromatous patients buried at an ancient leprosarium and from controls obtained from medieval cemeteries and present-day donors. Analysis showed an association of DRB1*15:01 allele with predisposition to lepromatous leprosy in medieval Europeans compared to contemporary and medieval controls. They also evaluated the genome of strains of *M. leprae* detected in the bones of leprosy patients by high-throughput sequencing and constructed a phylogenetic tree using additional medieval and modern strains.

The authors brings original and relevant data, as the conclusion that past epidemics, such as leprosy, could have influenced the frequency of alleles associated with modern chronic diseases; however some points need to be reviewed and/or better explored to become

adequate for publication. Major and minor appointments to be considered or answered by authors are presented below.

Major appointments

1. Assuming that the authors have produced sequence/genotyping data for the entire MRC/HLA locus, the major question that rises upon reading the manuscript is to why only data for DRB1 has been analyzed. This is particularly relevant given that there are numerous reports showing association between leprosy and several HLA variants, including classes I and III. Did the authors test other HLA loci for association? If not, what were the criteria for selecting DRB1 as the only target for analysis? Of note, the literature references presented to support DRB1*15:01 allele as a major candidate is somewhat incomplete.

In our study, we focused on DRB1*15:01 as it is – within the HLA region – the strongest known leprosy risk factor in very diverse populations. We have described this more clearly in the manuscript now. In addition, we have targeted DRB1*15:01 as the SNP rs3135388 is known to be a reliable marker for DRB1*15:01. Because of power considerations and to avoid multiple testing, we selected only this locus as we had a very restricted number of skeletons (affected and unaffected) and too few ancient HLA genotypes for a sound statistical analysis. Thus, we did not test other loci for association with LL.

We have now cited 5 publications (Ref. 5-9) that refer to the association of DRB1*15:01 and leprosy. Moreover, the article by Jarduli et al. 2013 is a review paper that cites the most important case-control studies focusing on the HLA region and leprosy.

2. Both in the abstract and the conclusion the authors make inference about a DRB1*15:01-DQB1*06:02 haplotype; it seems that molecular data for this haplotype has been generated (ref. table S19); however, no results are presented and/or discussed. Please clarify.

We agree that this data is relevant and thank the reviewer for pointing this out. We have now included this information in the generally expanded results section on HLA allele calling:

“Interestingly, in all 13 cases where DRB1*15:01 could be called (representing 12 individuals, including one homozygote), we also found the allele DQB1*06:02, suggesting strong linkage disequilibrium between these alleles at the two loci (supplementary information, Table S20). These two alleles indeed define a DRB1-DQB1 haplotype that is still found in modern Europeans at considerable frequencies.”

3. A large segment of the manuscript is dedicated to a study of *M. leprae* molecular epidemiology/phylogeny that is not mentioned in the title/abstract, thus resulting in an overall report that lacks uniformity and clarity of the main message to be conveyed. In short, the feel is of two almost independent papers in one. Perhaps it would be advantageous to concentrate on the *M. leprae* genomic data just as a necessary tool to prove infection and genomic stability, leaving complementary analysis – such as phylogeny – to a subsequent report. Please consider.

As recommended by the reviewer and the editor, we have restructured and refocused the manuscript. The emphasis is now on both ancient HLA association study and *M. leprae* genomics. This is mainly reflected in the new abstract and the discussion. The *M. leprae* genomic data is used to define the *M. leprae* DNA-positive individuals as cases, to point out the stability of the *M. leprae* genomes over the last 1000 years and to assess aDNA damage patterns. The phylogeography and phylogeny are also discussed. In addition, we investigated a possible link between the observed *M. leprae* strains and HLA alleles. We kept the original title as it conveys the most important finding of our study. However, if deemed necessary by the reviewers/editor we are happy to change the title.

4. The authors insist in classifying the ancient leprosy individuals studied as “LL”, a clinical form of disease defined as such only in 1966 in a classic paper by Ridley & Jopling. Also, the authors use expressions such as “severe” to define disease status – all this without having access to clinical/epidemiological/microbiological data necessary for such inferences. How did the authors conclude for cases being of “severe” leprosy, compatible with the LL form as described by R&J? Why not BL, for example?

According to Ridley & Jopling (1966) only the severe form of leprosy (LL) leads to bone lesions. This is supported by numerous other studies (e.g. Andersen et al. 1994, *Int J Osteoarchaeology* 4: 21-30; Lastoria & Milanez Morgado de Abreu 2014, *An Bras Dermatol* 89: 205-218). Specific bone lesions are routinely used as diagnostic criteria for the classification of LL in archaeo-anthropological material (e.g. Inskip et al. 2015, *PLoS One* 10: e0124282; Andersen et al. 1994). The classification of our cases as LL is described in detail by Boldsen 2001 (*J Phys Anthropol* 115: 380-387).

5. In the supplementary data the authors indicate excavating more skeletons than the ones apparently used; throughout the text, different numbers are presented as samples analyzed (79? 69? 68?). The same inconsistency is observed for controls. Could the authors please clarify?

We did not mention the total number of excavated individuals per site. The number of samples that were collected for analysis in our study are mentioned in the Supplementary Information (page 3-4) and in the Table S1 (St Jørgen n=85; Ribe n=42; Revshale n=45; Tirup n=57; Rathausmarkt n=79).

In the manuscript, we have re-written the Results section to facilitate the readability of the text and to highlight which sample sizes were used for which analysis (e.g. rs3135388 data for the association study, DRB1*15:01 data in the HLA profiling).

6. It is not clear in the discussion and conclusions which ways the DRB1*15:01-DQB1*06:02 haplotype and the results obtained could influence the frequency of genetic factors associated with modern inflammatory diseases.

We have now addressed this point more clearly in the discussion, outlining that leprosy should have led to a frequency decline of this allele and haplotype. We continue further that since it is

still very common in modern populations, likely other fitness effects are also associated with this allele, counteracting the deleterious effect of leprosy in medieval times.

7. Please justify the choice of minimum 10x coverage for de novo assemblies once larger coverage is usually required for this kind of approach.

10 x is the threshold used in our former aDNA study (Schünemann et al. 2013, Science) and was shown to be sufficient. A 10 x coverage already allows researchers to reliably call variants in bacterial genomes. In ancient genome studies, it is quite exceptional to obtain such a high coverage from shotgun data.

8. Considering the use of the Sanger sequencing technique, why rs3135388-T was used for sequenced instead of actual DRB1*15:01?

The allele nomenclature for classical HLA alleles (e.g. DRB1*15:01) refers to unique protein sequences of the domains that shape the antigen-binding groove of the HLA molecule. In case of the DRB1 molecule, the binding groove is shaped by exon 2 of the DRB1 gene, which spans 267 bp to code for 89 amino acids. A target length of 267 bp (plus necessary flanking sequences for primer annealing) is far beyond the average DNA fragment size of ancient samples. And the exon itself is far too variable to design reliable primers inside the coding sequence. Direct sequencing of the exon sequences for HLA genes is thus only possible from modern high-quality DNA that is little fragmented.

Even modern SNP-based GWAS approaches have no SNP marker inside these variable exons, because the variability would lead to unreliable hybridization success. Classical HLA genotypes are then inferred by imputation, which is exactly what we are doing here (inferring presence of HLA allele from presence of nearby linked SNP allele). Note, however, that we are also verifying the linkage between this SNP allele and DRB1*15:01 in a subset of individuals for which both genotyping assays were successful.

We have now included a brief explanation of this point in the manuscript:

“Direct Sanger sequencing of the relevant DRB1 exon is not possible from aDNA, because the length of the exon far exceeds the average DNA fragment size of ancient samples and the polymorphic exon sequence precludes the use of intra-exon primers.”

Minor appointments

1. Could the authors elucidate if the genomes were or not enriched? Information in the manuscript (line 77) and supplementary information (page 17) are contradictory.

The *M. leprae* and human genomes were generated after shotgun HTS without enrichment (p. 6, paragraph *M. leprae* genome analysis and metagenomic screening). The HLA region was enriched by capture before HTS (p. 6 - 8, paragraph *HTS-based HLA typing*).

2. For the sake of precision, please substitute continents by actual endemic countries (lines 58 and 59);

We have included some countries as example for each geographical region with the highest total number of new cases per year (page 3).

3. Please include the meaning of “IBD” abbreviation also in the manuscript;

We have explained the abbreviation IBD (identical by descent) in the manuscript (page 11).

4. Please review the use of the word ‘major’ as used in line 40;

It has been replaced by the word strong.

5. If acceptable by the journal definitions, please consider including a ‘methods’ subtopic. Also, it could be helpful to have flow chart describing the experimental design included as supplementary information;

We have rewritten the paper to increase the readability of the manuscript and included a well-structured methods section that describes the workflow of the experiments (p. 13-18).

6. Please check if there is any more recent (than 1944) reference for leprosy diagnosis by bone analysis;

We have replaced the 1944 reference by a more recent publication (Ref 1 and 10).

7. The word ‘loci’ needs to be in italic, please adjust. On the same note, please check if all the occurrences of *M. leprae* in the text are in italic;

Thank you - we have corrected the writing of Latin words throughout the manuscript.

8. It is not clear in the manuscript what was the biological material used to obtain the present-day controls. Please comment;

The present-day data are provided by the DKMS, a non-profit organization, recruiting potential volunteer donors for hematopoietic stem cell transplantation. Donors have been routinely typed at high resolution for HLA-A, -B, -C, -DRB1, -DQB1, and -DPB1 upon registration. DKMS also records self-assessed parentage, categorized by country of origin. Mainly saliva sample are used as biological material (details see Supplementary information, page 21).

9. Line 165: please consider substituting the word ‘victims’ by ‘patients’ or similar;

We have changed the wording as suggested.

10. Many references in the supplementary information do not correspond with the subject cited in the text (e.g. 'arm positions' (3) and 'osteological analyses' (16)); please perform a careful inspection of all references in the supplementary information.

Thank you, unfortunately, during reformatting the order of references was shifted. We have now carefully reviewed and corrected the reference list.

11. Please include p values and OR in figure 4.

We have included p values and OR in Figure 2 (former Figure 4).

12. S1 table: a) sex determination by bones and molecular tests are divergent, please justify the choice; b) please provide a properly legend for sex, especially for 'C' and '?'; c) Instead of the green tag in TagSNP, we suggest adding a column containing the number of PCRs for each case; d) what is the meaning of 'n' and 'T' in the TagSNP column? e) why there are little information about age and sex in the 'Revshade' tab?

As suggested by the reviewer, we have made the following changes to Table S1:

a) We have reported the results of both methods in order to be transparent. Inconsistent results in sex determination (osteology vs. molecular typing) are commonly observed. The sex determination based on osteological measurements can be imprecise, especially if the material is not completely preserved.

b) A proper legend is now provided (? = osteological sex determination was not possible).

c) We have added a column with the number of PCRs for each case.

d) We have provided a proper legend (n= no data available, T = T allele)

e) Only the samples that were positive for the genetic analysis were analyzed by the anthropologist.

13. Could the authors please provide the following adjustments, as listed below?

- Tables S12 and S14 have the same title;

- Legend for 'NA' is missing in table S20;

- In table S18 a 'm' is missing in the word 'number'; also, please provide a legend for the colors and a description of the meaning of 'pi hat' and 'nsnps';

- Table S7: legend for 'nc' is missing.

We thank the reviewer for pointing out these mistakes; they have been corrected in the supplementary.

14. It would be interesting to include the commands for human sequencing analysis in the supplementary information, as done for the bacterial DNA analysis.

The commands for the human sequencing analysis have been included in the supplementary (pages 16 and 17).

Reviewer #1 (Remarks to the Author):

The revised manuscript reads very well and the results are convincing.

I stand corrected with regards to the HLA-DRB1*15:01 association test and thank the authors for the clarification.

Reviewer #3 (Remarks to the Author):

The authors have addressed adequately all the points raised by the reviewers and in our opinion is now suited for publication.

Point-by-point response to the referees

REVIEWERS' COMMENTS:

Reviewer #1 (Remarks to the Author):

The revised manuscript reads very well and the results are convincing. I stand corrected with regards to the HLA-DRB1*15:01 association test and thank the authors for the clarification.

Thank you very much.

Reviewer #3 (Remarks to the Author):

The authors have addressed adequately all the points raised by the reviewers and in our opinion is now suited for publication.

Thank you very much.